# Diversity of bacterial small RNAs drives competitive strategies for a mutual chaperone

Jorjethe Roca [1], Andrew Santiago-Frangos [2,3] & Sarah A. Woodson [1✉]

Hundreds of bacterial small RNAs (sRNAs) require the Hfq chaperone to regulate mRNA expression. Hfq is limiting, thus competition among sRNAs for binding to Hfq shapes the proteomes of individual cells. To understand how sRNAs compete for a common partner, we present a single-molecule fluorescence platform to simultaneously visualize binding and release of multiple sRNAs with Hfq. We show that RNA residents rarely dissociate on their own. Instead, clashes between residents and challengers on the same face of Hfq cause rapid exchange, whereas RNAs that recognize different surfaces may cohabit Hfq for several minutes before one RNA departs. The prevalence of these pathways depends on the structure of each RNA and how it interacts with Hfq. We propose that sRNA diversity creates many pairwise interactions with Hfq that allow for distinct biological outcomes: active exchange favors fast regulation whereas co-residence of dissimilar RNAs favors target co-recognition or target exclusion.

[1] T. C. Jenkins Department of Biophysics, Johns Hopkins University, 3400 N. Charles St., Baltimore, MD 21218, USA. [2] CMDB Program, Johns Hopkins University, 3400 N. Charles St., Baltimore, MD 21218, USA. [3] Present address: Department of Microbiology and Cell Biology, Montana State University, Bozeman, MT 59717, USA. ✉email: swoodson@jhu.edu

The functions of cellular RNAs are influenced by their interactions with RNA-binding proteins (RBPs), which may exchange or remodel the RNAs over their life cycle[1,2]. When RBPs are limiting in number, their RNA ligands compete for binding[3–12]. This competition is often thought to depend on the order of recruitment, or the binding energetics. However, RBPs usually contain several RNA-binding domains, which not only confer stability to the complexes, but also allow a single RBP to interact with multiple RNAs[13–15]. Moreover, these multiple interaction sites can facilitate RNA exchange, enabling the continual remodeling of RBP complexes.

Bacterial small RNAs (sRNAs) regulate metabolism and stress response[16] by base pairing to a complementary region of an mRNA and altering its translation or stability[17]. sRNA regulation is facilitated by Hfq, a hexameric ring-shaped protein capable of simultaneously binding sRNAs and mRNAs[18]. This simultaneous binding is accomplished by distinct surfaces that recognize U-rich and A-rich motifs in the RNA substrates of Hfq[19]. Class I sRNAs interact with the proximal face and outer rim of the Hfq ring, whereas their mRNA targets bind the distal face of the ring. Class II sRNAs bind to the proximal and distal faces and base pair with targets that interact with the rim[20].

The copy number of sRNA and mRNA targets of Hfq generally exceeds the pool of free protein. As a result, sRNAs must compete for binding to Hfq, which stabilizes them against turnover. This adds to the sum of interactions that contribute to sRNA regulation within a cell. Pioneering studies established that sRNA overexpression decreases regulation by endogenous sRNAs, because of competition for binding to Hfq[6,7,21–23]. Additionally, one sRNA can regulate multiple targets, and multiple sRNAs can regulate a single target, resulting in a highly interconnected regulatory network dependent on Hfq availability[24]. Indeed, a recent report proposed that competition among targets for Hfq determines the subset of mRNAs that are regulated by sRNAs[11].

Although sRNAs bind Hfq tightly, potentially delaying their exchange, yet regulation occurs in just a few minutes upon sRNA expression[25]. Thus, it has been argued that sRNAs cannot passively wait for other RNAs to dissociate from Hfq before binding themselves[26,27]. Indeed, Fender and co-workers[26] showed that sRNA off-rates increase with competitor RNA concentration, suggesting that RNAs actively "cycle" on Hfq. Fast regulation could also be achieved if competitors do not need to displace a resident from Hfq to engage with their targets, that is, both sRNAs reside on the protein. However, active exchange and coexistence between two RNAs bound to a chaperone have not been directly observed, to our knowledge. Consequently, our understanding of how sRNAs gain access to Hfq is still limited.

The mechanism of competition is important for understanding how individual RNAs perform within a network of similar RNA ligands. Previous work has shown that competition performance is affected by sRNA structure, with the class II sRNAs being more proficient at displacing other sRNAs from Hfq[28–30]. Thus far, competition performance is not predicted by the binding affinities, association, or dissociation rates of individual RNAs[31]. However, these properties are rarely measured in the presence of other RNAs. Thus, we still do not know which RNA features affect competition for Hfq or for other RNA chaperones.

In this work, we directly observe sRNA competition for Hfq in real-time, using a new method of immobilizing single Hfq hexamers for single-molecule total internal fluorescence (smTIRF) microscopy. Our results show that resident sRNAs rarely dissociate from Hfq before a competitor sRNA binds (passive competition). Instead, resident sRNAs are rapidly displaced after they clash with an incoming challenger (active competition). Unexpectedly, the results also show that two sRNAs may reside on an Hfq hexamer for more than 20 s (stable coexistence).

We propose that stable coexistence may support new forms of sRNA regulation and may alter the engagement with mRNA targets. Importantly, we show that the mechanism of competition depends on the identity of the sRNA pair, not the resident or competitor alone, explaining the diversity of outcomes. Our findings on competition suggest nuances in regulation that could guide the design of novel synthetic circuits[32–34] and stimulate research on active RNA exchange on other RBPs.

## Results

**Single-molecule observation of sRNA exchange on Hfq.** To study how sRNAs compete for Hfq, we devised a smTIRF microscopy assay, in which the Hfq hexamer is biotinylated on one of its CTDs (BioHfq; see Methods). The biotinylation tag had only a minor effect on Hfq's function in sRNA regulation (Supplementary Figs. 1 and 2) and was present in BioHfq at less than 1 tag/hexamer, on average (Supplementary Fig. 3). BioHfq was complexed with a Cy3-labeled sRNA (the resident) and the complexes were immobilized on a passivated microscope slide treated with Neutravidin (Supplementary Fig. 4a, b). sRNA competition was studied by adding Cy5-labeled sRNAs (the competitors) to the slide and recording the colocalization of Cy3- and Cy5-labeled molecules with immobilized Hfq complexes in real time (Fig. 1a). In this experimental approach, a competitor sRNA can attempt to bind and replace a resident sRNA already occupying Hfq. To understand the means of competition, we tested DsrA (DA), a moderately competitive class I sRNA, and ChiX (CX), a strongly competitive class II sRNA, in various combinations (Supplementary Fig. 5 and Supplementary Table 1).

To determine how easily a resident is replaced by a competing sRNA, we counted the number of resident sRNA·Hfq complexes before adding the competitor and 5 min afterward. First, we observed that the level of resident remaining on Hfq decreased as the competitor's concentration was increased (Fig. 1b). This observation recapitulated the concentration dependence of sRNA competition measured previously by ensemble biochemical methods[26,28,29,31].

Second, we observed that the degree of competition depended on the particular resident and competitor sRNA pair. The most replacement occurred when class II ChiX sRNA challenged the class I DsrA sRNA pre-bound to Hfq (H-DA vs CX; 15% resident remaining at 5 nM competitor). The least replacement occurred when DsrA searched for a place on Hfq pre-bound to ChiX (H-CX vs DA; 87% resident remaining at 5 nM competitor). These results agreed with previous reports that ChiX sRNA is a powerful competitor for Hfq[6,12,28], and with the notion that sRNAs can compete more robustly for Hfq when, like ChiX, they interact with both its proximal and distal faces[29].

Third, we found that when a class I or class II sRNA competed against itself (H-DA vs DA or H-CX vs CX), an intermediate number of residents were displaced (42% resident remaining at 5 nM competitor). Interestingly, the degree of competition was similar for both homotypic sRNA pairs. This suggested that the extent of exchange is limited when sRNAs compete for the same binding surfaces on Hfq.

**sRNAs depart Hfq through active competition.** To study the competition process, we observed the binding and dissociation of single sRNA molecules in real-time for 5 min after adding competitor sRNAs. Figure 1c shows the colocalization of fluorophore-labeled sRNAs with an immobilized Hfq over time. In this representative trajectory, several Cy5-labeled competitors transiently bind Hfq before one of them forms a stable complex. This stable binding correlates with a loss of signal from the resident Cy3-sRNA.

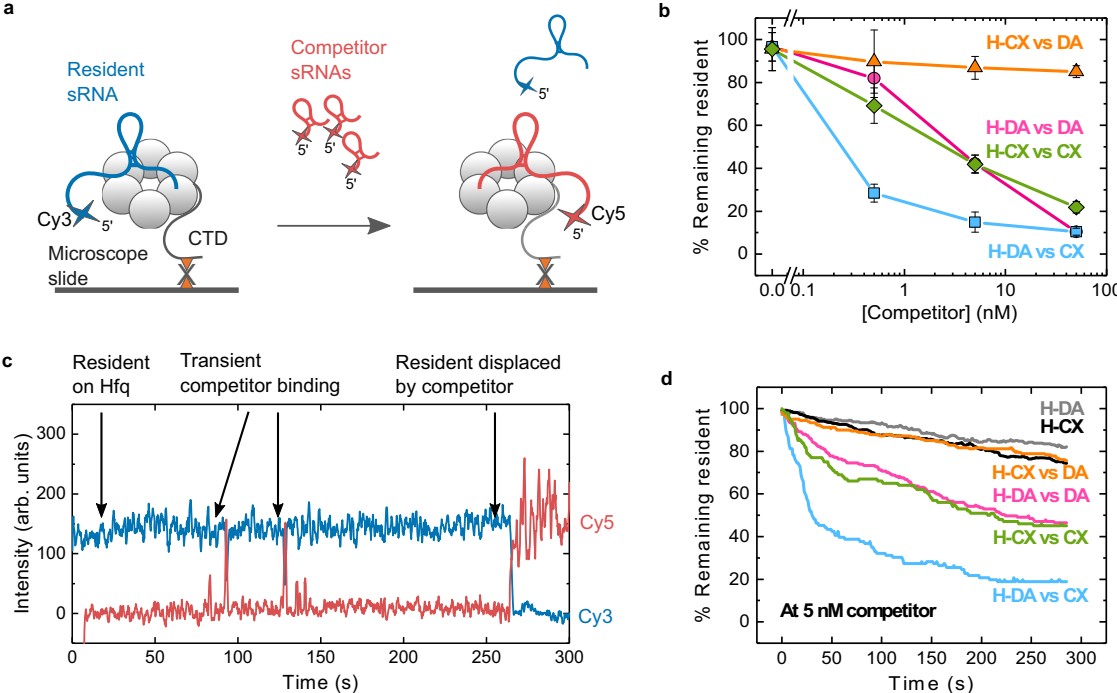

**Fig. 1 Single-molecule assay for real-time sRNA exchange. a** Biotinylated Hfq (BioHfq) hexamers were complexed with Cy3-labeled resident sRNAs and immobilized on a passivated microscope slide treated with NeutrAvidin. Sparse biotinylation ensures that most hexamers have only one immobilized CTD. The addition of Cy5-labeled competitor sRNA triggers displacement of resident Cy3-sRNA over time (see also Supplementary Fig. 4b). **b** Percent remaining resident sRNAs 5 min after addition of competitor sRNA, as a function of competitor concentration. Symbols represent the mean and s.d. of 10 different fields of view (FOV) from two independently prepared samples. Solid lines are a visual guide and do not represent a mathematical model. H: Hfq, DA: DsrA, CX: ChiX. **c** Representative single-molecule trajectory depicting the displacement of a resident sRNA from Hfq following competitor binding. **d** Representative plot (out of two replicates) showing the percentage of resident remaining on Hfq as a function of time, following the addition of 5 nM competitor sRNA. The number of active Cy3 molecules in a single 100 ms frame was counted every 10 frames (1 s) and divided by the total number of Cy3 molecules at the start of the movie. H-DA (gray) and H-CX (black) represent the dissociation in the absence of a competitor. Source data are provided as a Source Data file.

To characterize the aggregate kinetics of resident displacement, we calculated the percentage of residents remaining on Hfq from 0 to 5 min following the addition of 5 nM competitor sRNA (Fig. 1d). The results showed that displacement is the largest and fastest when DsrA (H-DA) is challenged by ChiX (CX), in agreement with the sRNA concentration dependence (Fig. 1b). By contrast, when ChiX was challenged by DsrA (H-CX vs DA), very little ChiX was removed, comparable to control experiments in the absence of a competitor (H-DA and H-CX). H-DA vs DA and H-CX vs CX resulted in intermediate levels of exchange with comparable displacement rates.

Our single molecule assay can distinguish passive competition, in which a resident sRNA spontaneously dissociates from Hfq before a competitor binds, from active competition, in which the resident dissociates after a competitor sRNA binds to the same Hfq hexamer. Strikingly, the resident sRNAs rarely dissociated from Hfq spontaneously during a 5 min movie; this passive mode accounted for ≤ 17% of sRNA exchange (Supplementary Fig. 6a). The levels of active vs. passive competition depended on the resident/competitor pair (Supplementary Fig. 6b), and active competition correlated with the amount of resident displaced (Fig. 1b, d). These results indicated that sRNAs mainly depart Hfq when actively displaced by other RNAs.

**Resident sRNAs impede the binding of incoming sRNAs.** Active competition has two steps: binding of a competitor, and displacement of the resident. Differential labeling of resident and competitor sRNAs allowed us to independently dissect the role of each of these steps in sRNA exchange. We hypothesized that both

steps would depend on the sRNA identities, because the competition outcome depends on which sRNA resides on Hfq when the competitor arrives. For example, when DsrA competes with itself, it is as good a competitor as when ChiX competes with itself, although ChiX prevails over DsrA when they compete against each other (Figs. 1b, d, and Supplementary Fig. 6b).

To assess whether a resident RNA impairs access of a competitor, we determined the time needed for competitor sRNAs to bind empty Hfq or Hfq loaded with a resident ($t_{bind}$; Fig. 2a). The cumulative fraction of bound competitor vs. time yielded time constants $\tau_{bind}$ for sRNA association, related to $\tau_{on}$ (Fig. 2b, c, Supplementary Fig. 7 and Supplementary Table 2). When Hfq was unoccupied, ChiX bound Hfq faster than DsrA (60% of CX events vs. 21% of DA events associated with $\tau_{bind} < 10$ s), consistent with ChiX's ability to interact with two surfaces of the Hfq hexamer. When Hfq was occupied by either DsrA or ChiX, however, binding slowed down, showing that either resident restricts access to Hfq.

This restriction clearly depended on the identity of the pre-existing sRNA·Hfq complex, because $\tau_{bind}$ for a given competitor differed depending on the resident. For example, DsrA bound Hfq slightly faster when ChiX was present than when DsrA was present (46 s for H-CX vs 71 s for H-DA; Supplementary Fig. 7e and Supplementary Table 2). This result can be rationalized if some ChiX molecules happen to only interact with the distal face of Hfq, leaving the proximal face open for DsrA to occupy. Conversely, ChiX associated with Hfq faster when DsrA was the resident (78% of events with $\tau_{bind} = 6$ s for H-DA and 60% with 15 s for H-CX; Supplementary Fig. 7f and Supplementary

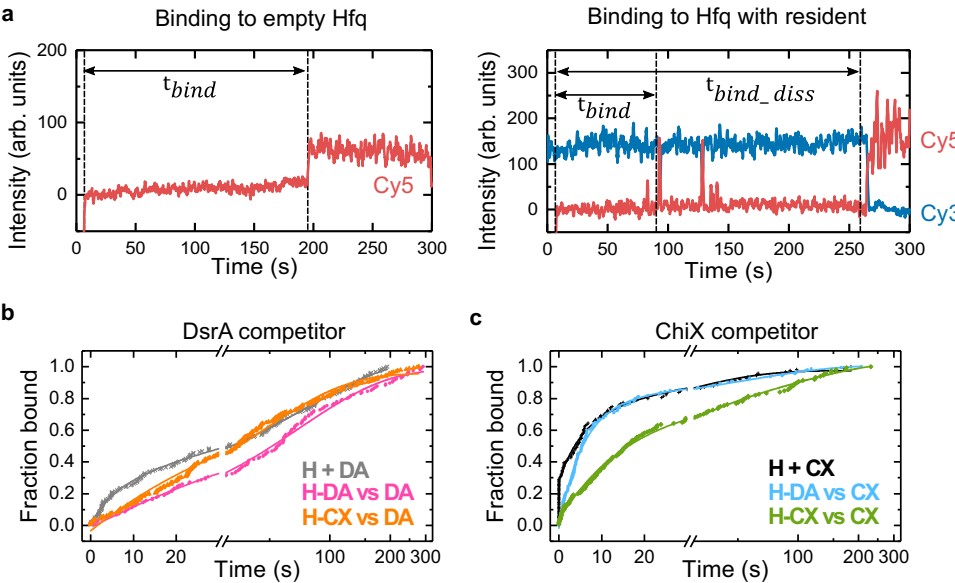

**Fig. 2 A resident on Hfq impedes competitor binding. a** Representative single-molecule trajectories showing a competitor binding to empty Hfq (left) or Hfq loaded with a resident sRNA (right). The association time is measured from the increase in Cy5 background to the first colocalization ($t_{bind}$) or displacement event ($t_{bind\_diss}$). **b, c** Cumulative fraction of competitor bound to empty Hfq (gray, black) or Hfq·resident complexes (colors) when the competitor is **b** DsrA or **c** ChiX. H: Hfq, DA: DsrA, CX: ChiX. The number of $t_{bind}$ events included in each plot are: $(H + DA) = 186$, $(H-DA \text{ vs } DA) = 154$, $(H-CX \text{ vs } DA) = 235$, $(H + CX) = 124$, $(H-DA \text{ vs } CX) = 154$, $(H-CX \text{ vs } CX) = 151$ for two independent experiments. To analyze the multi-phase binding kinetics, rate equations containing one, two or three exponentials were fitted to the data to obtain kinetics parameters and their errors (Supplementary Fig. 7 and Supplementary Table 2). Source data are provided as a Source Data file.

Table 2). Again, this can be rationalized if ChiX quickly engages the distal face of Hfq when the proximal face is unavailable.

Despite these differences, a short binding time did not correlate with more successful competition. For example, DsrA binds Hfq-ChiX complexes faster than Hfq-DsrA complexes, yet DsrA displaces relatively little resident ChiX (Figs. 1b, d, and Supplementary Fig. 6b). Thus, even though a resident sRNA affects competitor access to Hfq, active competition is not mainly determined by the speed of competitor binding. Therefore, we wondered if other features of the binding events correlated with subsequent displacement of the resident sRNA.

To elucidate this, we determined the competitor binding times only for those instances in which the resident departed ($\tau_{bind\_diss}$; Fig. 2a, Supplementary Fig. 7 and Supplementary Table 2). Although competitor binding to Hfq seemed modestly slower when the resident was displaced, compared to all binding events, this difference was not significant (K-S test). Thus, displacement is mainly determined by the competitor's ability to gain access to Hfq in the presence of an obstructive resident.

**Most sRNA exchange is fast.** To better understand what leads to sRNA removal after a competitor begins to invade Hfq's RNA binding surfaces, we determined the time elapsed from the moment of competitor binding until the departure of the resident ($t_{diss}$; Fig. 3a, b). A probability density plot of $t_{diss}$ revealed that most sRNAs are cleared from Hfq within 20 s after a competitor interacts with the protein (Fig. 3c). A maximum likelihood analysis of the probability density yielded three dissociation lifetimes (Supplementary Fig. 8 and Supplementary Table 3) that varied between resident/competitor pairs. The slowest category ($\tau_{diss3} = 58 - 133$ s) constituted a minor portion of displacement events (13-22%), relative to the faster categories ($\tau_{diss1} = 0 - 0.5$ s and $\tau_{diss2} = 4 - 13$ s).

In contrast with our results, previous ensemble studies reported very slow displacement lifetimes of $\sim 40 - 5,000$ s[26,28,29,31]. This difference may be due to the inability of ensemble experiments to

detect fast exchange and separate competitor binding from resident displacement, and the inability of smTIRF to measure lifetimes beyond the 300 s observation window of our movies. Nevertheless, both approaches showed that sRNA dissociation is faster in the presence of competitors.

**sRNA clash leads to active displacement.** The results above suggested that sRNAs are displaced because they hinder the binding of a competitor sRNA (Fig. 2, Supplementary Fig. 7 and Supplementary Table 2). Therefore, we hypothesized that rapid displacement arises from a clash between the resident and the incoming challenger, which would be more prominent when the sRNAs compete for the same binding sites. Indeed, maximum likelihood analysis indicated that during self-competition (H-DA vs DA and H-CX vs CX), $\tau_{diss1} = 0$ s, suggesting that displacement was quicker than the 0.2 s time resolution of our measurements (Supplementary Fig. 8 and Supplementary Table 3).

To determine whether an sRNA clash also explains the fast displacement of DsrA by ChiX, which can bind the distal and proximal face of Hfq, we challenged DsrA·Hfq complexes with truncated ChiX sRNAs. Full-length ChiX efficiently displaced DsrA from the proximal face, leaving only 15% of resident molecules bound to Hfq (Figs. 1b, d and 3d). Interestingly, a truncated ChiX unable to bind the proximal face of Hfq was also effective, leaving only 18% resident DsrA bound (ChiXΔtail; Fig. 3d and Supplementary Table 1). By contrast, a significantly shorter truncation of ChiX unable to reach the proximal face or rim of Hfq drastically reduced competition, with 79% of the resident DsrA remaining bound (ChiX_dist; Fig. 3d). These results suggested that some overlap of the sRNA binding sites – in this case, on the rim – is needed for active exchange.

**A single competitor is sufficient for resident displacement.** For some molecules, we noticed a stepwise increase in Cy5 intensity, suggesting that more than one competitor RNA had loaded onto Hfq before the resident was displaced (Fig. 3e). We calculated the

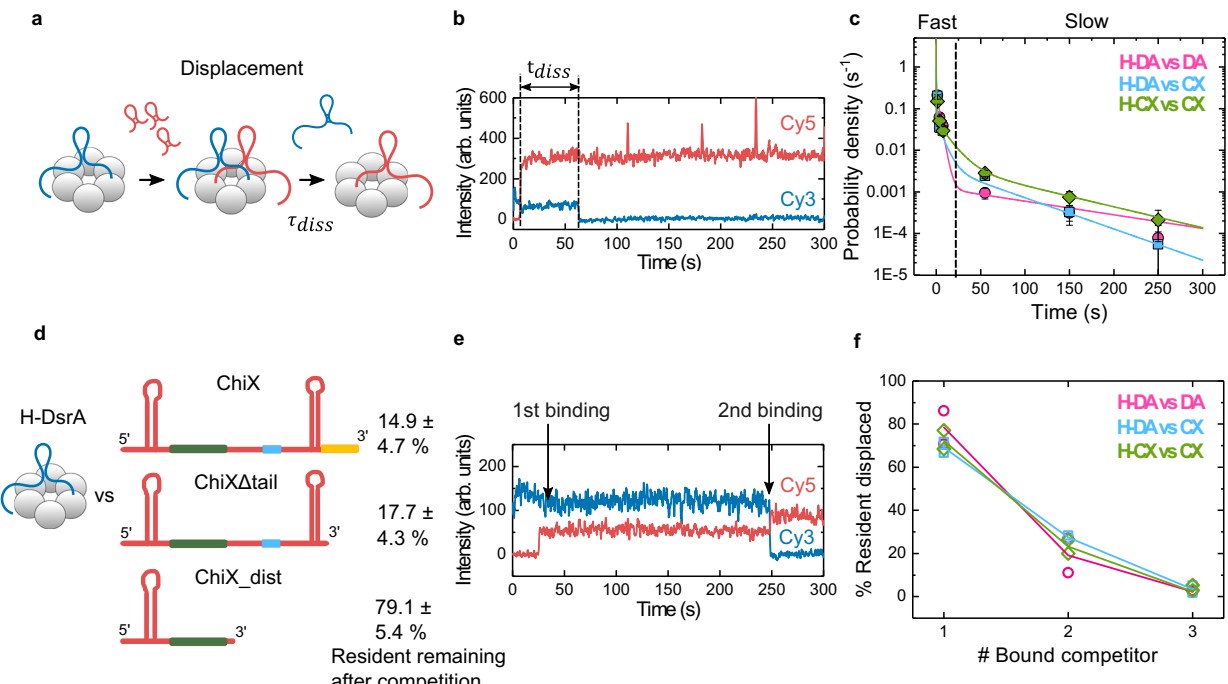

**Fig. 3 Clash of RNAs on Hfq leads to fast displacement. a** Two-step sRNA exchange during active competition. **b** Representative trajectory showing the resident displacement time ($t_{diss}$) after a competitor binds to Hfq. **c** Probability density histogram of $t_{diss}$. Solid lines represent a maximum likelihood fit to a three-exponential function. Error bars represent the variance in a binomial distribution[55]. The vertical dashed line at 20 s separates fast from slow displacement events. H: Hfq, DA: DsrA, CX: ChiX. The number of $t_{diss}$ events for each plot are: (H-DA vs DA) = 128, (H-DA vs CX) = 182, (H-CX vs CX) = 94 from two independent experiments. See also Supplementary Fig. 8 and Supplementary Table 3. **d** Percentage of DsrA resident remaining 5 min after addition of 5 nM ChiX, ChiXΔtail (nts 1-79) or ChiX_dist (nts 1-41). Values and errors were obtained as in Fig. 1b. Secondary structures were predicted using Mfold[59]. Yellow, blue, and green lines represent segments interacting with the proximal, rim, and distal faces of Hfq, respectively. **e** Representative single-molecule trajectory showing a resident displaced by two competitors. **f** Percentage of displaced residents arising from binding of one, two, or three competitor molecules. The symbols show the results of two independent experiments. Solid lines are a visual aid only. Source data are provided as a Source Data file.

number of competitors that bound to Hfq before each resident molecule was displaced (Fig. 3f). The majority (69–79%) of residents were displaced by a single competitor. Displacement by two competitors was less common, accounting for 19–28% of the events. Rarely (2%), we observed up to three competitors attached in the presence of the resident. Therefore, although possible, sRNAs are infrequently cleared by the interaction of multiple competitors and instead are driven off by the binding of a single competitor.

**Resident and competitor sRNAs stably coexist on Hfq.** Although many sRNAs dissociate from Hfq soon after a competitor sRNA arrives, sometimes the resident and competitor sRNAs cohabit Hfq for several minutes before one or the other sRNA dissociates (Fig. 4a). This observation raised the question of whether certain sRNA features increase the likelihood that two RNAs can be accommodated on the same Hfq hexamer.

To address this question, we determined the dwell times for events with two or more sRNAs (coexistence times, $t_{co}$), whether the resident or the competitor dissociated first (Fig. 4a). Maximum likelihood analysis of all co-existence events revealed distinct populations, in which one sRNA dissociated within 20 s ($t_{co}$ < 20 s) or the sRNAs coexisted on Hfq for more than 20 s ($t_{co}$ > 20 s) (Supplementary Fig. 9 and Supplementary Table 4; see Methods regarding the 20 s threshold). The long-lived events, which we refer to as stable coexistence, could conceivably impact

sRNA half-lives, target engagement, and regulation, by retaining two sRNAs on Hfq for a more extended period.

The prevalence of stable coexistence varied among the resident/competitor pairs tested (Fig. 4b). When both sRNAs competed for the same face of Hfq (H-DA vs DA and H-CX vs CX), stable coexistence was infrequent (16–17% of events), indicating that two identical sRNAs cannot readily occupy the same surface of Hfq. By contrast, when the resident and competitor recognize different Hfq surfaces (H-DA vs CX and H-CX vs DA), the proportion of stable coexistence was significantly higher (39–43% of events). Thus, sRNAs that can bind to different faces of Hfq are more likely to share the same hexamer. Interestingly, the coexistence lifetimes of the two homotypic complexes (~70 s) or the two heterotypic complexes (~40 s) were similar, regardless of the sRNAs involved, suggesting that stable co-binding arises from an analogous organization of the two sRNAs on Hfq. Additionally, we hypothesize that stable coexistence of sRNAs in the same face of Hfq can only occur for very tight and favorable RNA configurations on Hfq, resulting in longer lifetimes for homotypic complexes; for sRNAs positioned in opposite faces this condition for binding could be more relaxed.

These observations of stable complexes with two sRNAs led us to hypothesize that residents and competitors can organize on Hfq with minimal overlap. To ask how many sRNAs can occupy Hfq simultaneously, we calculated the percentage of Hfq molecules with stable coexistence events involving two, three or four sRNAs (Fig. 4c). We found that about half the time

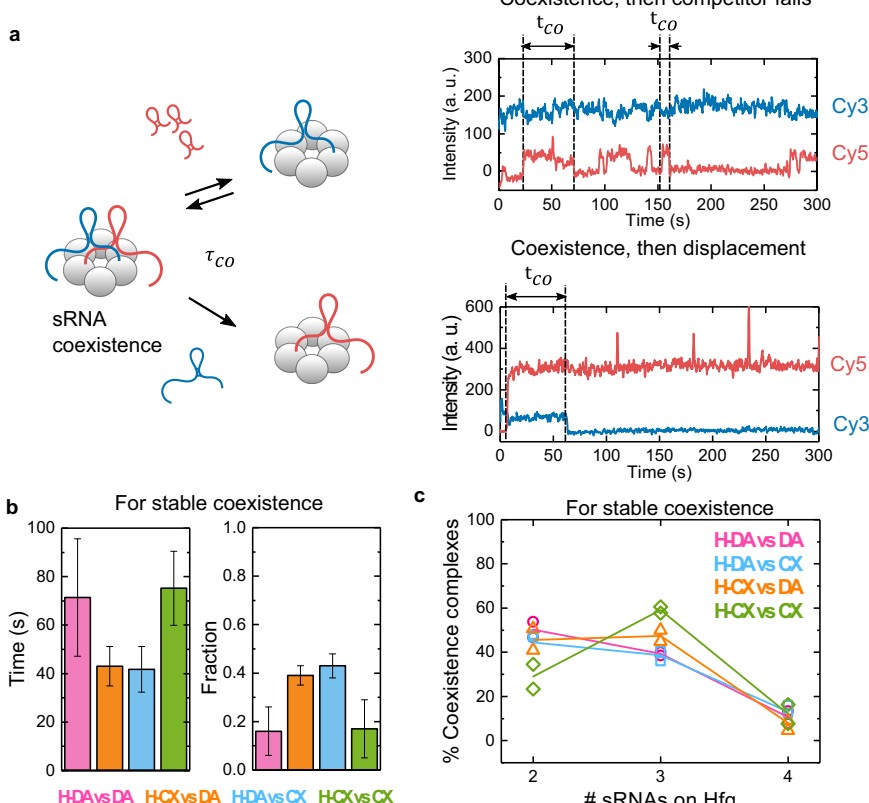

**Fig. 4 Some RNAs stably coexist on Hfq. a** Representative trajectories showing resident and competitor coexistence times ($t_{co}$), which end when either the competitor (top) or resident dissociates (bottom). Note that a single Hfq hexamer can form both long and short-lived complexes. **b** Characteristic lifetimes and fractions of the longest-lived populations obtained from maximum likelihood fits to the probability densities of $t_{co}$ for all co-existence events (Supplementary Fig. 9a). Error bars represent the variance in a binomial distribution (see also Supplementary Fig. 9b and Supplementary Table 4). **c** Hfq accommodates more than one sRNA. Percent stable complexes with two, three or four sRNAs (resident + competitor) over the total number of Hfq molecules showing stable coexistence. The symbols show the results of two independent experiments. Solid lines are for visual aid only. Source data are provided as a Source Data file.

(29–50%), two sRNAs occupied Hfq, around the other half (39–59%), three sRNAs did, while coexistence of four sRNAs was less frequent (8–13%). These results were strikingly different from events leading to fast sRNA exchange, in which just a single competitor was sufficient to displace the resident sRNA (Fig. 3d).

**RNA·Hfq contacts predict active competition.** Our results above indicate that the outcome of a competition experiment depends on the structures of both sRNAs and how they interact with Hfq. To understand further how sRNA structure affects the competition for Hfq, we performed competition experiments with an additional class I sRNA RydC and a class II sRNA CyaR (Fig. 5). Together with DsrA and ChiX, the four studied sRNAs are expected to form a wide range of interactions with Hfq, contacting the proximal, rim, and distal regions of the protein (Supplementary Fig. 5). According to the number of nucleotides in the Hfq-recognition motifs (Supplementary Table 5), RydC is expected to make the fewest number of interactions, followed by DsrA, and then the class II sRNAs CyaR and ChiX (Fig. 5, cartoons in heat maps, Supplementary Fig. 5).

For all sRNA pairs tested, passive competition accounted for fewer than 17% of sRNA exchange events (Fig. 5 top and Supplementary Table 6). Thus, passive competition, in which sRNAs spontaneously dissociate from Hfq before another RNA can bind, is unlikely to influence regulation at the cellular level, particularly in an environment crowded with other nucleic acids.

The active competition was most common for residents with fewer nucleotides that interact with Hfq or competitors with more interacting nucleotides (Fig. 5 bottom and Supplementary Table 8), as reflected by frequent resident displacement (Supplementary Fig. 10 left and Supplementary Table 9). As anticipated, class I residents that mainly contact the proximal face of Hfq were, in general, more susceptible to displacement than class II sRNAs, leading to more exchange overall. For most sRNA pairs, the competitor·Hfq interactions correlated with the number of competitor·Hfq complexes formed by the end of the experiment (Supplementary Fig. 10 middle and Supplementary Table 10). An exception to this trend was the class II sRNA CyaR, which exhibited a mediocre level of active competition, especially when Hfq was occupied by ChiX, despite CyaR being able to interact with two faces of Hfq. This unexpected behavior was explained by poor CyaR binding when Hfq was already occupied by ChiX (~ 17%; Supplementary Fig. 10 right and Supplementary Table 11). In contrast, CyaR was able to displace the class I sRNA DsrA from Hfq (Fig. 5 bottom). Given that CyaR's rim and distal binding motifs are weaker than those of ChiX, it is possible that CyaR mainly competes for the proximal surface, causing the displacement of class I but not class II residents.

**Interactions with opposite Hfq faces encourage coexistence.** Next, we determined the likelihood that Hfq is simultaneously occupied by each sRNA pair for more than 20 s (Fig. 5 middle

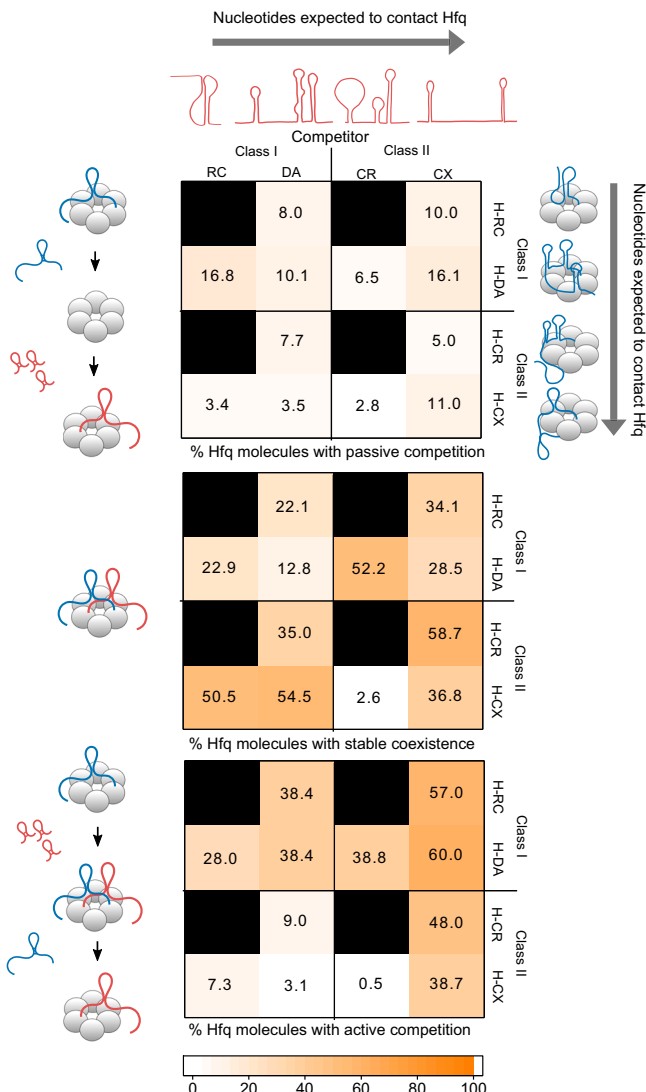

**Fig. 5 sRNA·Hfq interactions modulate competition strategies.** Heat maps depicting the percent of resident·Hfq complexes that experience passive competition (top), stable coexistence (middle), and active competition (bottom) for various resident and competitor combinations; H: Hfq, RC: RydC, DA: DsrA, CR: CyaR, CX: ChiX. Average and s.e.m of two independent experiments; see Supplementary Tables 6–8 and 12 for details. The cartoons represent the determined or predicted sRNA structures and their interactions with Hfq. The values in the three heat maps may not sum to 100% because some Hfq complexes do not experience any type of competition during the movie, and others exhibit both stable coexistence and resident displacement (active competition). Source data are provided as a Source Data file.

and Supplementary Table 7). Unlike the active competition, we found no clear trend between the possible number of sRNA·Hfq contacts and the likelihood of stable coexistence. Instead, stable coexistence was the least common for class I vs class I sRNAs (13–23%), perhaps because these sRNAs must interact with a limited area around the inner proximal pore of the Hfq hexamer. In contrast, when Hfq was initially bound by a class II sRNA, class I sRNAs often formed a stable complex (35–55% stable coexistence), reflecting the ability of the sRNAs to occupy opposite faces on Hfq. Interestingly, even though ChiX and ChiXΔtail easily displace DsrA from Hfq (Fig. 3d) and rarely stably co-exist, ChiX_dist, which can only bind the distal face,

commonly forms longer-lived complexes with DsrA. This supports the idea that co-residence on Hfq increases when the RNA interactions do not overlap (Supplementary Fig. 11).

## Discussion

RNA-binding proteins (RBPs) typically have several binding domains or surfaces[13–15]. As a result, the stabilities of RNA-protein complexes depend on the number of surfaces an RNA contacts, as well as the strength of each contact. Conversely, each RBP can usually recognize many RNAs, resulting in competition amongst similar RNAs for limited copies of shared proteins[3–12]. Our results show that binding to empty Hfq after passive dissociation of a previously bound RNA is rare, as previously proposed[26]. Instead, an incoming sRNA competes either by stably coexisting with its predecessor on Hfq, or by displacing the existing occupant (Fig. 6).

Although these exchange pathways coexist in real time, their frequencies depend on how the RNAs engage the protein: RNAs capable of forming many contacts with Hfq favor active competition (rapid exchange), while RNAs that interact with opposite faces of Hfq favor stable coexistence (with either RNA disengaging from Hfq) (Fig. 6). As a result, kinetic competition can produce a spectrum of biological outcomes, depending on the features of the sRNAs present in a cell at any given moment. The active competition allows rapid take-over of the Hfq pool, ensuring fast and efficient deployment of stress-induced sRNAs. As discussed below, this type of competition may suppress noise from spurious RNAs while enhancing the dominance of highly expressed sRNAs. Stable coexistence may allow for more nuanced regulation by protecting sRNAs from degradation, enabling co-regulation of a target or blocking target recruitment (Fig. 6).

Although sRNA exchange likely requires Hfq's multiple RNA binding surfaces, the displacement mechanism remains unknown. Our measurements of the binding kinetics suggest that active exchange occurs through a clash between a resident on Hfq and an incoming competitor. Although the resident impedes access by an RNA competing for the same surface of Hfq, subsequent displacement of the resident is fast. Thus, sRNA association is rate-limiting (Fig. 6). Only two studies have reported impaired association with an occupied Hfq hexamer[26,35] as opposed to empty Hfq[26,31,35–37]. Given that Hfq is likely always engaged with RNA in the cell[26], impeded binding to an occupied Hfq should be the most common scenario.

Although resident sRNAs often dissociate within the same movie frame as competitor binding ($\tau_{diss1}$ = 0–0.5 s), instances of slow dissociation ($\tau_{diss2}$ = 4–13 s) imply some rearrangement of the RNAs between arrival of the competitor and departure of the resident. In the cycling model[26], resident sRNAs are displaced when a competitor progressively invades the binding pockets of adjacent subunits, ultimately disengaging the resident from Hfq. Using this model, the authors estimated a resident dissociation rate of 0.06 s$^{-1}$ or a lifetime of 16.7 s, which is similar to our $\tau_{diss2}$. Alternatively, a resident could be electrostatically repelled by the incoming RNA or dislodged by the flexible CTDs as they accommodate a second RNA. Further research will be needed to elucidate the mechanisms of RNA exchange on Hfq.

Unexpectedly, we observed that two sRNAs sometimes stably coexist on Hfq, even though they are not expected to base pair. This stable coexistence is more frequent when the two sRNAs bind different Hfq surfaces (Fig. 5 middle). Decreased overlap of RNAs on Hfq could increase coexistence by reducing the likelihood of a clash, or by allowing the two sRNAs to diffuse into a stable arrangement after initial binding. We noted that coexistence is most prevalent when the resident sRNA binds the distal face of Hfq (class II). By contrast, coexistence is rare when both sRNAs must bind the proximal face (class I). Stable co-binding of

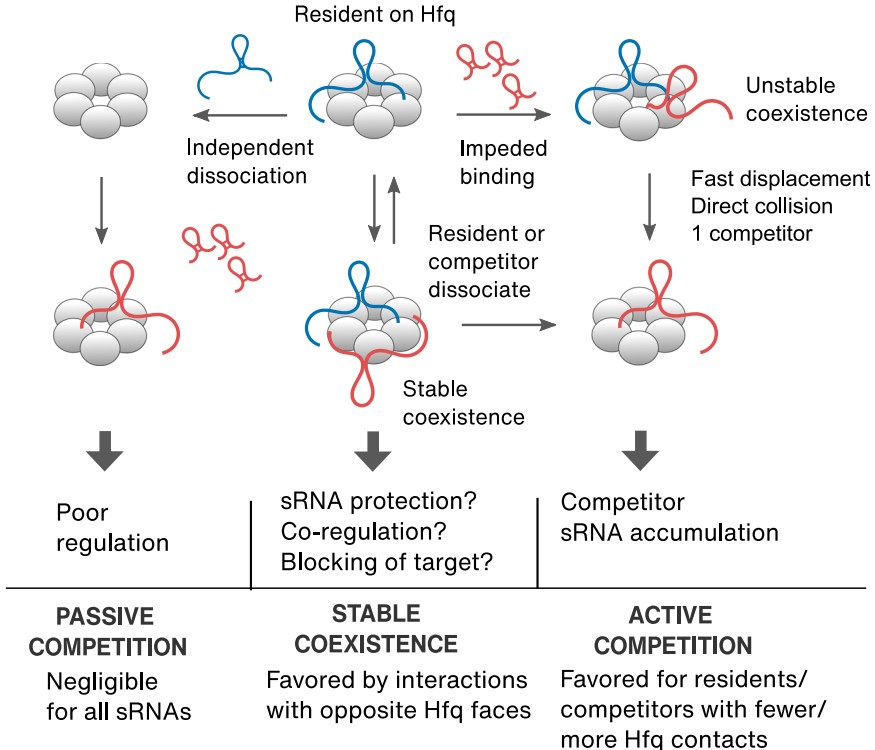

**Fig. 6 Model of sRNA competition strategies.** The variation in natural sRNA·Hfq interactions produces a range of outcomes depending on the structures of each sRNA pair (Fig. 5). Passive competition (left) is uncommon. Stable coexistence arises from prolonged co-binding of two sRNAs (middle), and fast displacement arises from a clash between competitor and resident (right).

two class I sRNAs may be discouraged by the difficulty of accommodating them both around the central pore of Hfq, or the propensity of the CTDs to displace RNAs from the proximal face[30,38].

Stable coexistence may explain certain unexpected patterns of sRNA competition in cells. Previously, it was reported that overexpression of RydC sRNA increased the expression of a target of ChiX, possibly by competing with ChiX for binding to Hfq[6]. This result was unexpected since ChiX has consistently been shown to be a superior competitor for Hfq[12,28,30]. Our experiments show that RydC is not good at displacing ChiX, but instead frequently forms a stable ternary complex on Hfq together with ChiX (Fig. 5 middle). We hypothesize that the presence of RydC on the rim of Hfq impairs the binding of ChiX's targets. Thus, one biological effect of stable coexistence would be decreased regulation by blocking the recruitment of targets.

Conversely, the arrangement of two sRNAs on the proximal and distal faces of Hfq may mimic the assembly of sRNA·Hfq·mRNA ternary complexes during normal regulation. In this scenario, base pairing between the two RNAs would lend additional stability to the ternary complexes, which might otherwise exchange with incoming RNAs. Recently[39], we found that partially base-paired ternary complexes have a lifetime of 2–4 s, whereas ~59% of class I annealed complexes lasted ≥90 s[39], which is about twice the lifetime of stable sRNA coexistence (Fig. 4b).

If two sRNAs can stably occupy opposite faces of Hfq, opportunities for forming other noncognate complexes are likely abundant in the cell, which could sequester Hfq and hamper regulation[27]. Recently, Park et al.[12] found that an unspecified fraction of ternary complexes in vivo included sRNAs and non-cognate mRNAs, when the RNAs bind to opposing Hfq surfaces. Our results with truncated ChiXΔtail (Fig. 3d and Supplementary Fig. 11) suggests that competition for the rim of Hfq, a surface

prone to non-specific RNA binding[30], can diminish non-cognate ternary complexes and prevent Hfq sequestration. Yet, Hfq sometimes accommodates two or more non-complementary RNAs in a way that doesn't lead to dissociation. What distinguishes these stable non-cognate ternary complexes from unstable complexes remains to be studied.

In our work, RNA association with Hfq was most hindered when the protein was occupied by a similar sRNA, as in the case of competing class I vs class I sRNAs (Fig. 2). We propose that this obstruction would discourage the sporadic binding of lowly expressed RNAs and promote regulation by highly induced sRNAs, as happens during stress. We also observed that association was less impaired when the competitor could bind an empty region of Hfq, as is the case for the class II sRNA ChiX competing against Hfq occupied with a class I sRNA (Fig. 2). First, this would allow access to targets, which usually bind orthogonally to their cognate sRNA. Second, easier access to Hfq by class II sRNAs could enable the fast removal of class I sRNAs, consistent with the proposal that class II sRNAs act more as silencers than emergency responders[20]. As ChiX can also displace targets from Hfq[12,23], unimpaired association with an empty proximal face could also prioritize target displacement by class II sRNAs, enabling a faster return to normal after a stress response. Thus, the transition from normal to stressed states, and vice versa, could be controlled by Hfq access, which in turn depends on the RNAs that occupy the protein.

Similar to previous work, we found that the class II ChiX sRNA is a strong competitor, easily removing RNAs and resisting displacement from Hfq[28,30]. In keeping with this dominance, ChiX's targets were not significantly affected by overexpression of other sRNAs in vivo[6]. Since ChiX is not degraded together with its targets[40,41], how do other RNAs regain access to Hfq? First, ChiX degradation is stimulated by an RNA decoy under specific growth conditions[40,41], potentially making Hfq available to other sRNAs.

Second, we hypothesize that the competition from class I targets could displace ChiX and allow for regulation. It remains to be seen if targets can actually engage in such complexes. Alternatively, for certain concentrations, competition of sRNAs for the proximal and of mRNAs for the distal face could stimulate ChiX's displacement from Hfq.

Hfq, an RBP that actively exchanges its ligands, possesses features common to other RBPs, such as multiple RNA recognition surfaces, clusters of basic and acidic residues, and intrinsically disordered regions. These similarities raise the possibility that other RBPs also facilitate the exchange of their RNA substrates. For example, microRNAs compete for the RNA-induced silencing complex (RISC)[3–5], pre-mRNAs compete for the splicing machinery[8], and mRNAs compete for RBPs involved in RNA metabolism[9,10]. In general, it is not known whether competition in these systems is passive, involving recycling of empty protein, or active, involving direct displacement. One study raised the possibility that Growth Associated Protein 43 (GAP-43) mRNA displaces β-actin mRNA from the Zipcode binding protein 1 (ZBP1) in neurons[42]. It was also noted before[31] that the dissociation rate of RNAs bound to the bacterial Rho factor increase with increasing concentrations of chase RNA[43,44]. Thus, active exchange may be an understudied mechanism explaining how RBPs capture their prospective ligands.

Our work has elucidated sRNA strategies to compete for the bacterial chaperone Hfq that highlight nuances in regulation when many RNAs are present, as occurs in the cell. These competition principles should aid in the design of Hfq-based synthetic circuits[32–34] with novel features, like target blocking. It still remains to characterize competition in the presence of cognate and non-cognate targets to better understand the interplay between RNAs during regulation. We hope that our work motivates the study of active displacement in other RNA-protein systems.

## Methods

**RNA preparation and fluorescence labeling.** DNA templates for sRNAs transcription were constructed by extension of overlapping DNA oligonucleotides (Thermo Fisher Scientific) using Q5 DNA polymerase (NEB, M0491L). One, two or three guanosines were added at the 5'end to improve transcription by T7 polymerase (Supplementary Table S1). sRNAs were transcribed in vitro using T7 RNA polymerase[45], except for the addition of 20 mM GMP. Transcription products were purified by PAGE and dissolved in water for immediate chemical modification and fluorescent labeling.

Fluorophores were attached to the sRNA 5' end in two steps[46] with some modifications: EDC (Pierce, PG82079) and ethylenediamine (Sigma-Aldrich, 195804-1006) were removed after the generation of a primary amine using centrifugal filters (3 K MWCO, Amicon Ultra 0.5 mL, Millipore Sigma, UFC5003) followed by ethanol precipitation of the RNA. For labeling, one package of Cy3 or Cy5 NHS mono reactive dye (Cytiva, PA23001 or PA25001) was dissolved in 60 μL DMSO (Invitrogen, D12345), and 30 μL of the dissolved dye was added to the RNA solution and incubated overnight at room temperature. Excess dye was removed using a Chroma TE-30 spin column (Takara Bio, 636069) followed by ethanol precipitation of the RNA. Excess free dye was not observed in purified labeled RNAs checked by denaturing PAGE. The labeling efficiency calculated from the ratio of dye to RNA concentration obtained from the absorption at 554 or 650 nm relative to 260 nm was 80–100% for all the labeled RNAs.

**Hfq expression plasmids.** To biotinylate Hfq for immobilization during single-molecule experiments, an AviTag sequence (GLNDIFEAQKIEWHE) was fused to the C-terminus of E. coli Hfq (EcHfq). The AviTag is recognized by E. coli's biotin ligase (BirA) which catalyzes the conjugation of a biotin to the tagged protein[47,48]. For over-expression, the tag sequence was added using around-the-horn site-directed mutagenesis of pET21b-EcHfq, with primer set 1 (Supplementary Table 13), resulting in pET21b-EcHfq-CAvi. For moderate expression, the coding sequence from pET21b-EcHfq-CAvi was amplified with primer set 2 (Supplementary Table 13), digested with SapI (NEB, R0569S) and subcloned into SapI-digested pD871 plasmid (Atum), to generate pD871-EcHfq-CAvi. A comparable plasmid expressing wild type E. coli Hfq, pD871-EcHfq, was prepared by PCR amplification with primer set 3 (Supplementary Table 13).

**Reporter assays for ChiX-chiP regulation.** To assess whether the C-terminal AviTag disrupted Hfq-dependent downregulation of chiP expression by ChiX sRNA, fresh colonies of DJS2689 (PM1205 lacI'::$P_{BAD}$-chiP-lacZ Δhfq::cat-sacB)[30], alone, or transformed with plasmids expressing untagged Hfq (pD871-EcHfq) or tagged Hfq-CAvi (pD871-EcHfq-CAvi) were used to inoculate 20 mL LB-Miller media supplemented with 0.001% rhamnose (plus 25 μg/mL kanamycin for transformed cells), and grown overnight (~16 h) at 37 °C, 200 rpm. The next day, 0.2 mL of each overnight starter culture was used to inoculate 20 mL of fresh LB-Miller media. All cultures were supplemented with 0.001% rhamnose and 0.004% arabinose, and cultures containing pD871 transformants were further supplemented with 25 μg/mL kanamycin. Cells were grown to an $OD_{600}$ between 0.7 and 0.8, and a β-galactosidase assay was performed (Supplementary Fig. 1).

**BioHfq expression and purification.** To decrease the impact of tagging and biotinylation on Hfq's function, Hfq hexamers containing around one modified CTD were prepared by co-expression of wild type Hfq and Hfq-CAvi. E. coli BL21(DE3) cells were co-transformed with plasmids expressing wild type Hfq (pET21b-EcHfq)[49] and Hfq-CAvi (pD871-EcHfq-CAvi). Transformed cells were grown in LB-Miller media supplemented with 100 μg/mL ampicillin and 25 μg/mL kanamycin at 37 °C until $OD_{600}$ = 0.5. Hfq and Hfq-CAvi expression was induced with 1 mM IPTG and 0.2 mM rhamnose (final concentrations), respectively. Biotin (100 μg/mL final concentration) was added with the inducers to promote the biotinylation of Hfq-CAvi monomers by endogenous BirA. Cells were grown for another 4 h at 37 °C and then collected by centrifugation and lysed, with the lysate treated and purified with a $Ni^{2+}$ affinity column[30]. The eluate from the $Ni^{2+}$ column was dialyzed into cation exchange buffer (50 mM sodium phosphate pH 6.5, 100 mM NaCl, 0.5 mM EDTA) and loaded onto a cation-exchange column (6 mL UNO S6, Bio-Rad, 720-0023) equilibrated in the same buffer. A buffer with pH 6.5 was chosen to discourage the binding of Hfq:Hfq-CAvi mixed hexamers with more than 3 tagged monomers (pI ≤ 6.4 estimated by ExPASy's compute pI tool[50]). The column was washed with cation exchange buffer and eluted with a linear gradient of 0.2-1 M NaCl. Desired fractions were pooled, dialyzed into HB buffer (50 mM Tris-HCl pH 7.5, 250 mM $NH_4Cl$, 1 mM EDTA, 10% glycerol), concentrated with centrifugal filters (3 K MWCO, Amicon Ultra, Millipore Sigma UFC8003) and stored at −80 °C. The obtained protein is referred to as BioHfq throughout the text.

**Assessment of BioHfq binding and competition by EMSA.** The impact of the modified CTDs on sRNA binding was assessed by comparing the affinity of DsrA for Hfq and BioHfq by EMSA. Native gel mobility shifts were obtained as reported previously[51], with the following modifications: 5'-$^{32}$P-labeled DsrA was 2 nM (final concentration), tRNA and tracking dyes were omitted, and reactions were incubated at room temperature for 30 min. A partition function for two independent sites was fitted to the measured fractions of protein-bound RNA[51]. Dissociation constants for Hfq and BioHfq were found to be similar (Supplementary Fig. 2a).

The effect of protein modification on competition performance was also tested by EMSA. $^{32}$P-DsrA (2 nM final concentration) was pre-bound to Hfq or BioHfq (5 nM final concentration) in TNK buffer (10 mM Tris-HCl pH 7.5, 50 mM NaCl, 50 mM KCl). After 30 min, 1 μL 10x TNK buffer or 0.01–1,000 nM unlabeled competitor DsrA were added to the reaction, and the incubation continued for another 30 min before samples were loaded in native 6% (w/v) polyacrylamide (29:1 mono:bis) gels. Gels were dried and quantified using ImageJ (1.53c). The bound protein-RNA fractions were fit with an empirical competitive model, as reported previously[30]. Competitor concentrations at which 50% of DsrA was displaced were in the same order of magnitude for Hfq and BioHfq (Supplementary Fig. 2b).

**Avi-tag Western blotting.** To experimentally determine the extent of tagging in BioHfq, we performed Western blots for the AviTag. 250 ng WT Hfq, 250 ng BioHfq (sparsely tagged), and 50 ng Hfq-CAvi (fully tagged) in 150 mM Tris·HCl pH 6.8, 1.5% SDS, 30% (v/v) glycerol were resolved in a 4-20% gradient Tris-Glycine gel (30 min at 200 V). Proteins were transferred to a nitrocellulose membrane using a semidry blotter and Towbin buffer (25 mM Tris, 192 mM glycine, 20% (v/v) methanol, pH 8.3) for 1 h at 65 mA. The membrane was blocked with 5% BSA in 1 × TBST (20 mM Tris, 150 mM NaCl, 0.1% Tween-20) for 1 h at room temperature, then incubated with primary antibody (0.5 μg/mL (1:1000 dilution) Avi Tag antibody, mAb, mouse; GenScript USA Inc., A01738-40) in 1× TBST, overnight at 4 °C. After three washes with 1× TBST, the membrane was incubated with the secondary antibody (2 μL antibody in 10 mL 1× TBST (1:5000 dilution); goat anti-mouse Alexa 594, Invitrogen, R37121) for 1 h at room temperature. The membrane was washed thrice with 1× TBST and allowed to fully dry, and then imaged in a Typhoon 9500 with excitation at 532 nm and emission collected with a long pass filter (>575 nm).

**Single-molecule competition experiments.** Single-molecule data were obtained using a home-built prism-type total internal reflection fluorescence (TIRF) microscope[52,53]. Cy3 and Cy5 fluorophores were excited with green (532 nm) and red (640 nm) lasers, respectively; emission intensities were collected with a 60X water immersion objective coupled to an EMCCD camera. Data recording was controlled using custom software (https://github.com/Ha-SingleMoleculeLab/Data-Aquisition). Short movies (50 frames, 100 ms/frame) were recorded with both

green and red lasers on. For long movies (3,000 frames, 100 ms/frame), the first and last second were recorded while exciting Cy3 and Cy5, while an alternating excitation scheme was used for the remainder of the movie to limit photobleaching.

Fluorophore-labeled sRNAs were refolded by heating at 90 °C for 1 min then cooling at room temperature for 10 min. Resident Cy3-sRNAs were pre-complexed with BioHfq by incubating 20 nM Cy3-sRNA with 10 nM BioHfq for 30 min in 1X TNK buffer (10 mM Tris-HCl pH 7.5, 50 mM NaCl, 50 mM KCl). Cy3-sRNA·BioHfq complexes were added to a DDS-passivated quartz slide pretreated with biotinylated BSA (0.2 mg/mL, Sigma, A8349-10MG), Tween-20 (0.2%) and NeutrAvidin (0.1 mg/mL, Thermo Scientific, 31000)[54] and incubated for 1-5 min. The slide was then washed with imaging buffer (10 mM Tris-HCl pH 7.5, 50 mM NaCl, 50 mM KCl, 4 mM Trolox (ACROS Organics, 218940010), 0.01% octaethylene glycol monododecyl ether (Nikkol), 0.8% glucose) supplemented with 165 U/mL glucose oxidase (Sigma, G2133-10KU) and 2 U RNasin Plus (PROMEGA, N2615). Cy3 spots were only observed when both BioHfq and NeutrAvidin were present on the slide (Supplementary Fig. 4a).

For competition experiments, short movies (~50 frames) were first recorded in different fields of view to determine the average number of Cy3-sRNA·BioHfq complexes before the competition. Immediately afterward, competitor sRNAs in the imaging buffer were flowed into the slide chamber during acquisition. The increase in the background in the Cy5 channel corresponded to the time of competitor addition. After the real-time competition experiment (~3,000 frames), short movies (~50 frames) were collected in different fields of view (FOV) to determine the average number of Cy3- and Cy5-labeled molecules after the competition. To minimize having two or more Hfq closely located, we aimed to balance a low spot density but with a significant number of molecules per experiment (~500 molecules).

**Single-molecule data analysis.** Single-molecule experiments were analyzed using the Imscroll software implemented in MATLAB[55]. Cy3-sRNA molecules were first selected as areas of interest (AOIs). AOIs were manually inspected to ensure single molecules were present; AOIs with two or more Hfq were not considered for further analysis. The intensity of the selected Cy3 AOIs was integrated over the entire length of the movie and then the AOI positions were mapped to the Cy5 channel to obtain the Cy3 and Cy5 intensity time trajectories[55,56]. Cy3 and Cy5 intensities for the same AOI were overlaid to visualize resident and competitors interacting with a single Hfq (Supplementary Fig. 4c).

To assess the contribution of photobleaching, the number of stably bound Cy3- and Cy5-sRNAs at the end of the 5 min flow movie in each channel were compared with the average number of fluorescent molecules in 5 different fields of view on the same slide (obtained from short movies). If these numbers were similar (<15% different), photobleaching was considered not significant, and such movies were analyzed further. Binding intervals for Cy3-labeled residents and Cy5-labeled competitors were obtained independently from Cy3 and Cy5 fluorescent traces of the same Hfq molecule[55,56]. Cy3 and Cy5 binding intervals were examined to determine the dwell times of Cy3- and Cy5-sRNAs colocalization (coexistence), Cy3-sRNA (resident) or Cy5-sRNA (competitor) (Supplementary Fig. 4c).

*Association of competitor sRNA.* The competitor association times with immobilized Hfq ($t_{bind}$) were determined from the time of arrival of the competitor sRNA solution (marked by a slight increase in Cy5 background fluorescence) to the start of the first Cy5 colocalization event (Fig. 2a and Supplementary Fig. 4c). Association times from two independent replicates were combined to construct cumulative density plots (fraction bound vs. time). To obtain the association kinetics parameters ($\tau_{bind}$ and its fraction) and their errors, rate functions containing one, two or three exponentials were fitted into the cumulative density plots, using OriginPro (2017). To measure the initial binding of a Cy5-sRNA to an empty Hfq, AOIs of Cy5-labeled complexes at the end of the movie were selected, and the intensity of these AOIs was integrated over the entire length of the movie to generate intensity traces. From these traces, only Cy5-sRNA·Hfq molecules that never colocalized with a resident Cy3-sRNA were used to construct cumulative density plots, as above.

*Displacement and coexistence times analysis.* Times for resident and competitor coexistence ($t_{co}$) were obtained from binding intervals showing colocalization (Fig. 4a and Supplementary Fig. 4c). Displacement times ($t_{diss}$) corresponded to colocalization intervals followed by intervals of competitor only binding (Fig. 3b and Supplementary Fig. 4c). Coexistence (or displacement) times from two independent replicates were combined to construct probability density plots of coexistence (or displacement) times; these distributions were fitted with double or triple exponential functions using maximum likelihood methods[55] to determine the characteristic times ($\tau_{co}$ or $\tau_{diss}$ and their fractions), with errors estimated by a bootstrap method[39,56].

*Stable coexistence analysis.* A 20 s threshold for defining stable coexisting resident/competitor pairs was established by examining probability density plots of coexistence times ($t_{co}$). These showed roughly one fast and one slow phase with a discontinuity around 20 s for all the sRNA pairs studied (Supplementary Fig. 9a). Based on this discontinuity in the kinetics, instances of co-residence with $t_{co} > 20$ s

were classified as stable coexistence events. In vivo studies have shown that the limiting step in target search is RNA association, with a reported apparent $k_{on} \sim 2 \times 10^5$ M$^{-1}$s$^{-1}$[57]. Considering ~10–60 copies of a given mRNA per *E. coli* cell[57,58] and a cell volume of ~0.5 μm$^3$, the concentration of an mRNA in the cell is ~ (30–200) nM, resulting in ~(25–150) s for the RNA to access Hfq. Thus, 20 s is a reasonable value for the minimum time an sRNA needs to remain on Hfq to regulate its targets.

**Statistical analysis.** Statistical details such as the number of experiments, molecules and events analyzed are detailed throughout the manuscript text, figure legends and supplementary information. Kolmogorov-Smirnov tests were calculated using Physics: Tools for Science (College of St Benedict, St John's University) at KS-test Data Entry (csbsju.edu)

**Reporting Summary.** Further information on research design is available in the Nature Research Reporting Summary linked to this article.

## Data availability
The raw data that support the findings of this study are available from the corresponding author upon reasonable request. The processed and analyzed data generated in this study are provided in the Source Data file.

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

## Acknowledgements

This work was funded by a grant from the National Institutes of Health [R35GM136351-01 to S.A.W.].

## Author contributions

J.R. prepared samples, designed and performed experiments, analyzed and interpreted the data and wrote the paper; A.S.F. designed and cloned Hfq-CAvi and tested its biological function; S.A.W. conceived the project and helped write and edit the paper.

## Competing interests
The authors declare no competing interests.
