## [Peer Review File · Nature Communications]

Title: Diversity of bacterial small RNAs drives competitive strategies for a mutual chaperoneREVIEWER COMMENTS

Reviewer #1 (Remarks to the Author):

The manuscript by Roca et al. uses single molecule fluorescence experiments to analyze the kinetics of binding of small regulatory RNAs (sRNAs) to the bacterial chaperone protein Hfq. Hfq serves as a platform, on which small regulatory RNAs reach their mRNA targets and form stable complexes with them. However, as there are numerous RNAs that can interact with Hfq, only a subset of these encounters on Hfq are likely to be productive. The manuscript focuses on this initial stage, which is the binding of sRNAs, in the absence of mRNA targets. The Authors aim to figure out how the alternative ways, in which different sRNAs form contacts with Hfq, determine the outcomes of their binding, i.e. whether the sRNA binding could ultimately lead to pairing with a regulated mRNA. Several aspects related to alternative binding modes to Hfq, sRNA exchange on Hfq, and competition for access to this protein have been subjects of previous papers. This manuscript for the first time presents a comprehensive picture of sRNA interactions with Hfq that takes into account the interdependence of sRNA regulatory outcomes. The Authors show that distinct Hfq binding specificities of sRNAs lead to either simultaneous or mutually exclusive sRNA binding on Hfq, and in this way determine the availability of Hfq for binding of other RNAs, such as mRNAs regulated by a bound sRNA. This elegantly planned and clearly written manuscript will be of interest for the general audience of Nature Communications and I recommend its publication.

Minor comments

- 1) In Fig. 5, sRNAs are ordered based on the evaluated strength of interactions with Hfq. However, the rationale to estimate relative strength of sRNA binding to Hfq on the analysis of their structures (presented on suppl. Fig 4) seems weak. A simple comparative measure would be to test their ability to outcompete each other, similarly as it was done in experiment shown on Figure 1, but including also RydC and CyaR (after taking into account 3' U tail lengths; see comment 3 below). The number of contacts between sRNA and Hfq does not need to be predictive of the binding strength, as binding of sRNAs to Hfq is often dominated by interactions between 3'U tail and the proximal face.
- 2) In Fig. 5 percentages describing what fraction of the Hfq-sRNA complex undergoes passive, soft, or active competition, often do not sum up: some are below 100% (HCX vs RC), and some are above (H-CR vs CX), please clarify.
- 3) In Suppl. Fig. 4 all sRNAs have equal length of 3'-U tail. Is it the natural length of the intrinsic terminator tail in all of them, or it has been arbitrarily selected, please clarify.
- 4) In the same figure, what was the rationale to have RydC sequence from *S. enterica* and others from *E.coli*, or are all four sRNA sequences from *S. enterica*?
- 5) Line 184 in results section, how did you categorize the events? Can it be simply explained?
- 6) On several plots values on axes are squeezed together, for example fig. 2d, e, f, Suppl. Fig. 2b
- 7) Paper by Ellis MJ et al (Mol Micro 2015) shows another proof of very efficient ChiX binding to Hfq in bacterial cells, which could be referenced.
- 8) Tagged Hfq is sometimes referred to as Hfq-CAvi and sometimes as BioHfq, e.g. Suppl. Fig. 1, please clarify.

9) What are the concentrations of labeled RNA, Hfq and BioHfq proteins, and RNA competitors on gels in Supplementary Fig. 2 ?

Reviewer #2 (Remarks to the Author):

In this manuscript, using single-molecule imaging and colocalization analysis, Roca et al measured the how bacterial small RNAs (sRNAs) compete for binding to Hfq proteins, which are limited in the cell compared to the abundance of the cellular RNAs, but are needed for sRNAs' stability and function. The work reveals different pathways for sRNA competitive binding using various combinations of resident sRNA and competitor sRNA: passive displacement, active competition, and an additional coexistence state; and elucidates important molecular mechanisms for sRNA-Hfq interactions. The experiments are carefully done and most parts of the manuscript are well written. I do not have major concerns, but have several specific points that I hope the authors can address. In addition, the presentation of the data/analysis in certain places needs to be clarified.

1. The authors analyzed the data from multiple angles, which is appreciable. However, it would be better to have a paragraph at early sections of the results to explain the relationship between different definitions and related parameters. The final model presents three pathways: passive competition, soft competition, and active competition. The definition of passive competition is straightforward. However, the rationale for defining the soft competition and active competition is not very clear. In Line 232, the authors define that the long-lived coexistence complexes (with $t_{co} > 20s$) are referred to as stable coexistence or soft competition, and the short-lived complexes (I assume with a $t_{co} < 20$) are defined as active competition. It is unclear why $t_{co} = 20$ is used for separate the two pathways, rather than using the decay time of the slowest population as a cutoff in corresponding cases (Fig S8). In the population of soft competition, does the resident RNA or the competitor RNA eventually dissociate within the imaging time? If so then why can't this be considered as "active competition" as well? I think there could be better definition of the active vs soft competition (I also like the term "coexistence state" more than "soft competition"): (1) by relating the t_{co} with t_{diss} : the populations in which the resident RNAs eventually dissociate are considered as active competition, whereas the populations in which both RNAs co-exist (no RNA dissociate) are considered as coexistence state. (2) by considering the "polarity" of the competition: the populations in which the resident RNA is always displaced by the competitor RNA are considered as active competition, whereas the population in which resident and competitor RNA have equal probability of dissociation after coexistence are considered as "coexistence state". It is not clear whether the current population analysis already consider these two criteria. I suggest the authors reanalyze the data in alternative ways to make the analysis, the definition and the model (Fig 6) more cohesive.

2. I am confused by the numbers presented in Fig 4b and Fig 5. Because the long-lived coexistence complexes (with $t_{co} > 20s$) are referred to as stable coexistence or soft competition, I assume these two terms are interchangeable and the percentages of these two should be the same. Line 236-240

states that when both sRNAs have the same binding face on Hfq (H-DA vs DA and H-CX vs CX), the coexistence was infrequent (16-17%), which is reflected in Fig 4b and Fig S8. However, the heat map in Fig 5 showed pretty significant percentage of soft competition for the cases of (H-CX vs CX), and are not consistent with the corresponding values showed in Fig 4b. Are the numbers presented in Fig 5 calculated in a different way? How would ChiX still form a stable coexistence complex with another ChiX on Hfq?

3. The data are in most of the cases fit with 3 populations, for both the departure time of the resident RNA (t_{diss}) and the coexistence time for the resident and competitor RNAs (t_{co}). Is there any interpretation of the nature of the three populations? The Hfq hexamers mixed with more than 3 biotin tagged monomers are washed during purification. Is there a way to quantify the actual distribution of biotinylated monomer in the sample? Could the populations with different t_{co} or t_{diss} be due to the sample heterogeneity?

4. The coexistence state of RNAs on Hfq has also been proposed by an earlier study by Part et al (eLife, 2021). While in that particular work, the coexistence state was observed between the class I sRNA and the cellular mRNA. The author should discuss any similarity or difference between the sRNA-mRNA-Hfq complexes observed in vivo and the sRNA-sRNA-Hfq complexes observed here. Related to this, it would be interesting to add one set of experiments to look at sRNA vs mRNA (non-matching) competition on Hfq.

5. It is interesting that the truncated ChiX unable to bind to the proximal face can still displace DsrA effectively. I am wondering whether similar phenomenon will be observed when using wild-type ChiX with Hfq carrying mutations at the proximal face (such as Q8A)?

6. Some negative fitting values are generated in the t_{diss} analysis (supplementary table S3). I am wondering for those case, whether 2-population fitting is good enough? Are they any model selection criteria used in the data fitting?

7. Line 204, I think Fig 3d should be cited in here instead of 5d?

Reviewer #3 (Remarks to the Author):

The manuscript “Diversity of bacterial small RNAs drives competitive strategies for a mutual chaperone” by Roca et al. deals with and significantly contributes to three important topics in the field of Hfq-mediated gene expression control in bacteria. First, it proposes a robust single-molecule approach to study the relationship between Hfq and its sRNA clients, which brings in information inaccessible to ensemble methods. As probably the most striking example of such new findings, one can now clearly see that sRNAs are capable of prolonged coexistence on the same Hfq hexamer, sometimes with surprisingly high stoichiometries and variable outcomes, which, put in the context of the timing of sRNA-dependent regulation in bacterial cells, will urge the community to reconsider the established view of

such mechanisms. This work also enriches and further rationalises the previously proposed division of Hfq-binding sRNAs into classes by adding a critical biochemical component - the way they behave when running against more or less apt competitors for a shared limiting resource (Hfq molecules). This elevates the definition sRNA classes from the original “one sRNA – Hfq – one target” paradigm to the more physiologically relevant “many sRNAs – Hfq – many targets” perspective, somewhat where the field is now clearly heading. Finally, this study sheds new light on the key phenomenon of the whole story - sRNA cycling on Hfq. Albeit known since already quite a long time and actively studied by several research groups, it remains elusive from the point of view of its driving forces and the extent to which we can predict the outcome of competition between two given sRNAs. Similarly to earlier studies, the authors come to the conclusion that bulk thermodynamic or kinetic parameters of individual sRNA-Hfq interactions cannot explain why some sRNAs compete for Hfq more successfully than others. By contrast, the topology of each sRNA-Hfq binding and the extent to which sRNAs are ‘flexible’ with regard to the choice of Hfq faces they engage seem to hold the key to this riddle. Specifically, the authors propose that clashing on the same Hfq surface will typically drive inevitable and fast sRNA exchange, whereas any possibility to occupy separate niches on the same Hfq hexamer will favour prolonged co-residence of several sRNAs (soft competition). At the same time, the first scenario is complicated by the poorer access of a competitor to the already occupied site, whereas the second looks like a sure way to slowly push the resident out by using unoccupied sites as a toehold for invasion. These somewhat opposed tendencies may indeed explain a good deal of what we currently know about the competition properties of class I and class II sRNAs. I also foresee that both the method and the proposed paradigm will stimulate similar research in other molecular systems involving binding of multiple RNAs to a shared protein.

While convinced by the data, I find that the authors may still improve their presentation by better explaining their analysis to non-specialists, providing details about certain controls, and clarifying or further supporting ambiguous wording/interpretations, as outlined in the following (minor) points.

1. It could be useful to better introduce the meaning of the multiple values in Supplementary tables. Why are they multiple in the first place? (One could probably add column subheadings like “ $\tau_{\text{bind_diss1}}$ ”, “ $\tau_{\text{bind_diss2}}$ ”, “ $\tau_{\text{bind_diss3}}$ ” and explain that they come from fitting two or three exponentials.) The term ‘amplitude’ may also be misunderstood by non-specialists (e.g., as ‘amplitude of the fluorescent signal’ instead of ‘proportion of molecules showing the specified behaviour’).
2. One technical point is the actual number of C-terminal tails per hexamer which are immobilised on the slide. This may be an important parameter, given that their movement significantly contributes to sRNA cycling on Hfq. Although the authors took multiple precautions to minimise the number of attachment-competent tails, do they have an idea about the actual Hfq/AviHfq ratio in their BioHfq prep? This could be done by SDS-PAGE or by western blotting.
3. Another technical issue that deserves a more detailed treatment is how the authors ensured that the spots they observed and quantified always corresponded to single Hfq hexamers and not to two (or more) closely located, microscopically unresolved complexes. This is obviously important in the context of the proposed multiple co-binding of sRNAs. Judging by the density of spots (Supplementary Fig. 3) and the high frequency of observed co-residency events (certainly higher than would be expected from by-chance colocalization; Fig. 4, Supplementary Fig. 8), I do not think that this factor may significantly

confound the data and affect their interpretation. But I feel that this issue should be properly discussed in the Methods section to exclude possible doubts.

4. I have a problem with term 'strength' as it is employed in the section "Strength of RNA-Hfq interactions predicts active competition" and in Fig. 5. What does it mean? It clearly is not the dissociation constant (the values reported for these sRNAs in literature are actually fairly close to each other and do not follow the order proposed in Fig. 5). To me this notion has more to do with the number of possible high-affinity contacts between each sRNA and different Hfq sites (as illustrated in Supplementary Fig. 4). This is not strength in a common biochemical sense (i.e., bulk affinity measured by an ensemble approach, without distinction of the underlying enthalpic and entropic contributions from different binding sites). I think it is important to be careful about wording in this section to avoid the already challenged interpretation that individual affinities (Kd) determine the outcome of sRNA competition for Hfq.

5. Fig. 3d and the corresponding main text: I was not aware that ChiX interacts with the rim. Could the authors comment on this?

6. To the same panel: I would replace "Resident left after competition" with "Resident remaining after competition" to avoid misunderstanding.

7. Inspired by the same panel and the last section of Results: in the case of ChiX_dist, does it show more soft competition than longer ChiX versions, or it loses the ability to displace DsrA altogether? Conversely, would a ChiX variant without the distal site-binding motif still be capable of soft competition?

8. L. 150: should not it be "60% of CX events", given the 15% initial burst mentioned in Supplementary Table 2?

9. Ll. 171-172: the difference between displacing and all binding events (Supplementary Fig. 6) seems to be small. Is it significant (e.g., by K-S test)?

10. L. 204: should be "Figs. 1b, 1d and 3d".

11. Ll. 241-244: the observation that the homotypic complexes exist longer than the heterotypic ones (70 s vs 40 s) seems to be paradoxical. Do the authors have an idea why it may be so?

12. L. 260-262: unless the authors have actual data or simulation results to support this assertion, I would tone it down to "Among them, RydC can be expected/seems to make the fewest number of interactions..."

13. Throughout the text: a better wording would be 'model/function is fitted into data' and not the other way around.

14. L. 611: This reference lacks further authors (E. Massé is not the only one on the paper).

15. L. 618: 'performance'.

16. Fig. 2f seems to be identical to Supplementary Fig. 6d.

17. L. 754 (Fig. 4c): the sentence "Additional sRNAs disrupt stable complexes" is not completely accurate since the coexistence of 2 or 3 sRNAs on Hfq seems to be equally likely.

18. L. 762 (Fig. 5): there are no representative trajectories here.

19. Supplementary Fig. 1: A supporting reference is needed for the statement "We chose this sRNA-mRNA regulatory pair..."

Alexandre Smirnov

REVIEWER COMMENTS

Reviewer #1 (Remarks to the Author):

The manuscript by Roca et al. uses single molecule fluorescence experiments to analyze the kinetics of binding of small regulatory RNAs (sRNAs) to the bacterial chaperone protein Hfq. Hfq serves as a platform, on which small regulatory RNAs reach their mRNA targets and form stable complexes with them. However, as there are numerous RNAs that can interact with Hfq, only a subset of these encounters on Hfq are likely to be productive. The manuscript focuses on this initial stage, which is the binding of sRNAs, in the absence of mRNA targets. The Authors aim to figure out how the alternative ways, in which different sRNAs form contacts with Hfq, determine the outcomes of their binding, i.e. whether the sRNA binding could ultimately lead to pairing with a regulated mRNA. Several aspects related to alternative binding modes to Hfq, sRNA exchange on Hfq, and competition for access to this protein have been subjects of previous papers. This manuscript for the first time presents a comprehensive picture of sRNA interactions with Hfq that takes into account the interdependence of sRNA regulatory outcomes. The Authors show that distinct Hfq binding specificities of sRNAs lead to either simultaneous or mutually exclusive sRNA binding on Hfq, and in this way determine the availability of Hfq for binding of other RNAs, such as mRNAs regulated by a bound sRNA. This elegantly planned and clearly written manuscript will be of interest for the general audience of Nature Communications and I recommend its publication.

Minor comments

1) In Fig. 5, sRNAs are ordered based on the evaluated strength of interactions with Hfq. However, the rationale to estimate relative strength of sRNA binding to Hfq on the analysis of their structures (presented on suppl. Fig 4) seems weak. A simple comparative measure would be to test their ability to outcompete each other, similarly as it was done in experiment shown on Figure 1, but including also RydC and CyaR (after taking into account 3' U tail lengths; see comment 3 below). The number of contacts between sRNA and Hfq does not need to be predictive of the binding strength, as binding of sRNAs to Hfq is often dominated by interactions between 3'U tail and the proximal face.

We appreciate the reviewer's recommendation for the ranking of sRNAs. However, competition results as shown in Fig. 1 only report on the resident remaining on Hfq and do not differentiate between residents displaced by competition or lost by passive dissociation. Additionally, we thought it would be informative to relate sRNA physical features with competition performance. We agree that the number of contacts may not predict binding strength, and we also recognize, as pointed out by Reviewer 3 (question 4), that the term "strength" is likely inappropriate for the feature we are considering. Thus, we have made revisions throughout the manuscript to remove the word "strength" in this context, and to clarify that we are referring to the number of nucleotides in the proximal, rim and distal recognition motifs that could contact Hfq (line 266 (title of section), lines 272-275, lines 280-281, line 299 (on discussion), Fig. 5, Fig. 6). We have also added a new table (Supplementary Table 5) detailing the sRNA ranking based on this criterion. The rank order of the 4 sRNAs is unchanged.

2) In Fig. 5 percentages describing what fraction of the Hfq-sRNA complex undergoes passive, soft, or active competition, often do not sum up: some are below 100% (HCX vs RC), and some are above (H-CR vs CX), please clarify.

The percentages are not supposed to add up to 100%. Some Hfq molecules do not undergo any of these forms of competition during the observation period, and thus the percentages sum to less than 100%. Similarly, since we define soft competition as long co-existence on Hfq, regardless of which sRNA remains at the end on the protein, some stable coexistence events are also active competition events. In this case, the percentages sum to more than 100%. We have noted this in the figure's caption (lines 828-829). Thank you for helping us clarify this point.

3) In Suppl. Fig. 4 all sRNAs have equal length of 3'-U tail. Is it the natural length of the intrinsic terminator tail in all of them, or it has been arbitrarily selected, please clarify.

They do not all have the same length. Due to a drawing error, ChiX is missing a U in Supplementary Fig. 4 (now Supplementary Fig. 5); this error has been corrected. We took our sequences from previous reports (DsrA: Małacka et al., *Biochemistry* 2015; ChiX: Schu DJ et al., *EMBO J* 2015, RydC: Dimastrogiovanni D, *eLife* 2014; CyaR: Lalaouna D et al., *NAR* 2017), which made it easier to compare our results with others. Of course, in nature, the position of transcription termination can vary by a few nucleotides.

4) In the same figure, what was the rationale to have RydC sequence from *S. enterica* and others from *E. coli*, or are all four sRNA sequences from *S. enterica*?

We chose RydC from *S. enterica* because its crystal structure in complex with Hfq is available, and thus we could see which nucleotides interact with Hfq (Dimastrogiovanni D, eLife 2014). We have incorporated this rationale into the caption of Supplementary Fig. 4 (now Supplementary Fig. 5). The two sRNAs are very similar. We added the sequence of RydC from *E. coli* to Supplementary Table 1 and we note that *S. enterica* RydC nucleotides that interact with Hfq are conserved in *E. coli* RydC.

5) Line 184 in results section, how did you categorize the events? Can it be simply explained?

The probability density of dissociation events was fitted using a maximum likelihood analysis. This fit produced the three characteristic lifetimes mentioned in the main text. We have clarified this procedure on line 183, new Supplementary Fig. 7, 8 and 9 and Supplementary Tables 2, 3 and 4. More information about the maximum likelihood analysis is found in the methods section “Single-molecule data analysis: Displacement and coexistence times analysis” and references therein.

6) On several plots values on axes are squeezed together, for example fig. 2d, e, f, Suppl. Fig. 2b

Thank you for bringing this issue to our attention. We have fixed the axes in Fig. 2, Supplementary Fig. 2 and Supplementary Fig. 6 (now Supplementary Fig. 7).

7) Paper by Ellis MJ et al (Mol Micro 2015) shows another proof of very efficient ChiX binding to Hfq in bacterial cells, which could be referenced.

Thank you for bringing this paper to our attention. We have added this citation in lines 47 and 388.

8) Tagged Hfq is sometimes referred to as Hfq-CAvi and sometimes as BioHfq, e.g. Suppl. Fig. 1, please clarify.

Hfq-CAvi refers to the Hfq protein sequence plus a C-terminal Avi tag. If pET21b-EcHfq-CAvi is the sole source of Hfq over-expression, hexamers will have the tag on all monomers (6 tags). For simplicity, this Hfq preparation is referred to as “Hfq-CAvi”. BioHfq refers to the protein preparation with a minimal amount of tagging, which was achieved by carefully calibrated co-expression of WT Hfq and Hfq-CAvi. We understand this nomenclature may be confusing, so we have clarified that “Hfq-CAvi” is fully-tagged where appropriate (line 504, new Supplementary Fig. 3).

9) What are the concentrations of labeled RNA, Hfq and BioHfq proteins, and RNA competitors on gels in Supplementary Fig. 2?

The concentration of labeled DsrA for binding and competition experiments was 2 nM. The protein concentrations for binding experiments were 0.25, 0.5, 0.75, 1, 2.5, 5, 7.5, 10 and 25 nM. The protein concentration for competition experiments was 5 nM. The concentrations of unlabeled competitor for competition experiments were 0.01, 0.1, 1, 10, 100 and 1000 nM. These details were added to the legend to Fig. S2.

Reviewer #2 (Remarks to the Author):

In this manuscript, using single-molecule imaging and colocalization analysis, Roca et al measured the how bacterial small RNAs (sRNAs) compete for binding to Hfq proteins, which are limited in the cell compared to the abundance of the cellular RNAs, but are needed for sRNAs' stability and function. The work reveals different pathways for sRNA competitive binding using various combinations of resident sRNA and competitor sRNA: passive displacement, active competition, and an additional coexistence state; and elucidates important molecular mechanisms for sRNA-Hfq interactions. The experiments are carefully done and most parts of the manuscript are well written. I do not have major concerns, but have several specific points that I hope the authors can address. In addition, the presentation of the data/analysis in certain places needs to be clarified.

1. The authors analyzed the data from multiple angles, which is appreciable. However, it would be better to have a paragraph at early sections of the results to explain the relationship between different definitions and related parameters. The final model presents three pathways: passive competition, soft competition, and active competition. The definition of passive competition is straightforward. However, the rationale for defining the soft competition and

active competition is not very clear. In Line 232, the authors define that the long-lived coexistence complexes (with $t_{co} > 20s$) are referred to as stable coexistence or soft competition, and the short-lived complexes (I assume with a $t_{co} < 20$) are defined as active competition. It is unclear why $t_{co} = 20$ is used to separate the two pathways, rather than using the decay time of the slowest population as a cutoff in corresponding cases (Fig S8). In the population of soft competition, does the resident RNA or the competitor RNA eventually dissociate within the imaging time? If so then why can't this be considered as "active competition" as well? I think there could be a better definition of the active vs soft competition (I also like the term "coexistence state" more than "soft competition"): (1) by relating the t_{co} with t_{diss} : the populations in which the resident RNAs eventually dissociate are considered as active competition, whereas the populations in which both RNAs co-exist (no RNA dissociate) are considered as coexistence state. (2) by considering the "polarity" of the competition: the populations in which the resident RNA is always displaced by the competitor RNA are considered as active competition, whereas the population in which resident and competitor RNA have equal probability of dissociation after coexistence are considered as "coexistence state". It is not clear whether the current population analysis already considers these two criteria. I suggest the authors reanalyze the data in alternative ways to make the analysis, the definition and the model (Fig 6) more cohesive.

In this work, we classified the competitive pathways according to the different ways a competitor could affect regulation through its occupancy of Hfq.

The cutoff $t_{co} = 20$ s was chosen because it divides most of the faster events (τ_{co1} , τ_{co2}) from the slowest population (τ_{co3}), as the reviewer suggests. However, we chose a threshold that could work for all of the sRNA pairs tested, so they could be evaluated with a common criterion. 20 s is 3-5 times τ_{co1} and τ_{co2} , so these faster processes have mostly decayed by this time. We have added a more detailed explanation of this choice to the Methods section "Stable coexistence analysis". In addition, we argue that these stable co-existence complexes are roughly comparable to the time needed for Hfq to regulate its targets, based on the annealing kinetics measured in *E. coli* by Fe et al (2015).

The reviewer is correct that the resident or competitor of a "stable" complex may ultimately dissociate sometime during the movie. When it is the resident that departed, these events are indeed considered "active competition", which comprises all events leading to sRNA exchange. We wonder if a sentence referring to short-lived events as characteristic of active competition gave the reviewer the idea that only short-lived events are included in the active competition analysis. We have deleted this sentence to avoid any misunderstanding.

The reviewer suggests some alternative classification schemes. We appreciate the reviewer's suggestions. However, we believe them to be less suitable for our purposes: Option (1) classifies intrinsic dissociation of the resident, in which the competitor is not responsible for the dissociation of the resident, as active competition; option (2) includes transient coexistence events in the soft competition category, which likely are not relevant for regulation during coexistence, as explained above.

We agree that soft competition may not be the best descriptor and we have changed this to stable co-existence in the revised manuscript as suggested. Additionally, to clarify things, we have now motivated the idea of the three competition pathways earlier in the introduction (lines 56-60 and 71-74) and defined and determined the three types of events in the new Supplementary Fig. 6). We have amended the text throughout in an effort to reduce confusion around the terms active competition and stable coexistence.

2. I am confused by the numbers presented in Fig 4b and Fig 5. Because the long-lived coexistence complexes (with $t_{co} > 20s$) are referred to as stable coexistence or soft competition, I assume these two terms are interchangeable and the percentages of these two should be the same. Line 236-240 states that when both sRNAs have the same binding face on Hfq (H-DA vs DA and H-CX vs CX), the coexistence was infrequent (16-17%), which is reflected in Fig 4b and Fig S8. However, the heat map in Fig 5 showed a pretty significant percentage of soft competition for the cases of (H-CX vs CX), and are not consistent with the corresponding values showed in Fig 4b. Are the numbers presented in Fig 5 calculated in a different way?

Thank you for calling our attention to this potentially confusing difference in the analyses in Fig. 4 and Fig. 5. Fig. 4b shows the fraction of **events** that are stable, while Fig. 5, middle panel shows the percentage of immobilized **Hfq hexamers** that experience a stable coexistence event. For the case of H-CX vs CX, ~40% of Hfq show at least one stable coexistence event during the observation period, but the fraction of coexistence events that are stable is just ~0.2. These results mean that ~40% of single Hfqs have many transient coexistence events until finally reaching a stable coexistence state, which is consistent with the individual trajectories. We have amended the text to clarify that

we refer to events in Fig 4. and lines 229, 231-234, and 240, that we refer to Hfq molecules (which can have more than one stable coexistence event) in line 301.

How would ChiX still form a stable coexistence complex with another ChiX on Hfq?

This could happen if one ChiX contacts the proximal face, while the other ChiX only contacts the distal face. For this to happen, the competitor ChiX should displace the resident ChiX from one face, but not the other.

3. The data are in most of the cases fit with 3 populations, for both the departure time of the resident RNA (t_{diss}) and the coexistence time for the resident and competitor RNAs (t_{co}). Is there any interpretation of the nature of the three populations?

As discussed in the manuscript, we believe that the shortest-lived population ($t_{\text{diss}1}$) corresponds to displacement due to RNAs clash (results' sections "Most sRNA exchange is fast" and "sRNA clash leads to active displacement"). As mentioned in the discussion (paragraph 4), it is possible that the second population ($t_{\text{diss}2}$) involves rearrangement of the RNAs after competitor binding. Finally, the longest-lived population ($t_{\text{diss}3}$) could correspond to exceptionally stable resident-Hfq-competitor complexes in which different RNAs interact independently with different Hfq surfaces. We argue that similar scenarios explain the three populations observed for coexistence, as t_{co} describes the interactions between the RNAs and Hfq regardless of which RNA remains or departs from the protein (Figure 4a). For example, $t_{\text{co}1}$ includes both fast resident displacement but also transient competitor binding. $t_{\text{co}2}$ could still describe populations for which the RNAs are rearranged and thus persist longer on Hfq, with eventually the resident or the competitor removed from the protein. Likewise, $t_{\text{co}3}$ includes stable complexes in which the RNAs occupy distinct Hfq surfaces (results' section "Resident and competitor sRNAs stably coexist on Hfq" and discussion, paragraph 5).

The Hfq hexamers mixed with more than 3 biotin tagged monomers are washed during purification. Is there a way to quantify the actual distribution of biotinylated monomer in the sample?

We performed additional gel mobility shift assays and a Western blot to evaluate the amount of Avi-tagged Hfq in the BioHfq preparation. These additional experiments are now presented in Supplementary Fig. 3. We were unable to observe super-shifts with streptavidin in the bulk protein sample, suggesting that the overall level of biotinylation is low. We next focused on quantifying the amount of Avi-tag in BioHfq. The change in charge-to-mass ratio conferred by each Avi-tag is expected to reduce the mobility of the complexes, as confirmed by controls with fully tagged Hfq-CAvi (new Supplementary Fig. 3a), providing a sensitive measure of tag incorporation. We were unable to detect a change in mobility of sRNA•BioHfq complexes, indicating that the Avi-tagged subunits are less than 10% of the total. We also attempted to detect Avi-tagged subunits by Western blotting with an AviTag antibody (new Supplementary Fig. 3b). Although the fully tagged Hfq-CAvi protein was readily detected by the antibody, BioHfq was undetectable, even when we loaded 5-fold more protein compared to the fully tagged Hfq-CAvi. This result also indicates that BioHfq contains $\leq 10\%$ Avi-tagged subunits. Using the Poisson distribution, and assuming 0.1 average frequency of tagged subunits, 90.48% hexamers are expected to be untagged, while 9.05% have one tag, 0.45%, two tags and 0.02%, three tags. Thus, of the hexamers that could be immobilized on the slide, only 4.9% could have more than one tag. This is a conservative estimate; the true frequency is likely lower. We have added a section to the Methods (Avi-tag Western blotting) and the legend to Supplementary Fig. 3 describing the procedure used to assess Hfq tagging.

Could the populations with different t_{co} or t_{diss} be due to the sample heterogeneity?

This is an important question. However, since the immobilized BioHfq likely consists of single-tagged hexamers, as discussed above, we do not think sample heterogeneity is the reason for multiple populations. Additionally, single hexamers can show a wide distribution of lifetimes that contribute to different states. This diversity of lifetimes is clear from representative traces, such as the one shown in Figure 4a, top right. We added a note about this to the legend for Fig. 4a. Thus, we believe the different populations represent different configurations of RNAs interacting with Hfq.

4. The coexistence state of RNAs on Hfq has also been proposed by an earlier study by Part et al (eLife, 2021). While in that particular work, the coexistence state was observed between the class I sRNA and the cellular mRNA. The author should discuss any similarity or difference between the sRNA-mRNA-Hfq complexes observed in vivo and the sRNA-sRNA-Hfq complexes observed here.

Thank you for reminding us of that particular result from Park et al. We have now incorporated it into our discussion (lines 372-378). In principle, we do not think there should be many differences between non-cognate complexes formed by two sRNAs occupying opposite Hfq faces and those formed by an sRNA and a non-cognate target mRNA. In this work, we show that non-cognate complexes are more common when the two RNAs do not have overlapping surfaces, as proposed by Park et al. (new Supplementary Fig. 11). However, we further show that even when the RNAs bind opposite surfaces, they may still compete for the rim, leading to RNA displacement. We hypothesize that this mechanism prevents Hfq sequestration by non-cognate complexes in the cell, and we expect that the numbers of such complexes in the cell are small. Since Park et al. could not distinguish mRNA-Hfq from sRNA-Hfq-mRNA complexes, and thus could not detect the non-cognate complexes directly, this question remains open. How two or more non-cognate RNAs (sRNAs only or sRNA and non-target mRNA) are structurally accommodated on Hfq is a subject for further study.

Related to this, it would be interesting to add one set of experiments to look at sRNA vs mRNA (non-matching) competition on Hfq.

We agree that this is a very interesting question, and it is one that we are presently investigating. However, given that the present manuscript is already quite long and complex, we think it better to save this topic for a different manuscript. Preliminary results show that non-matching complexes are uncommon and short-lived.

5. It is interesting that the truncated ChiX unable to bind to the proximal face can still displace DsrA effectively. I am wondering whether similar phenomenon will be observed when using wild-type ChiX with Hfq carrying mutations at the proximal face (such as Q8A)?

This experiment would be hard to do since DsrA binding to an Hfq proximal face mutant will be weak, yielding few Hfq-resident molecules. Even if this experiment can be done, we worry that the results will be confounding, since many resident dissociation events could arise from poor binding and not displacement by a competitor. It is for this reason that we employed the strategy of mutating ChiX rather than Hfq itself.

6. Some negative fitting values are generated in the t_{diss} analysis (supplementary table S3). I am wondering for those case, whether 2-population fitting is good enough?

No fitting values were negative; values <0 were not assigned in the maximum likelihood fit because they are faster than the time resolution of the experiment, as discussed in the section "sRNA clash leads to active displacement" and Supplementary Fig. 7 (new Supplementary Fig. 8). We realize our current nomenclature is confusing and have replaced these parameters with their upper bound, ≤ 0.2 , in Supplementary Table 3. Thank you for bringing it to our attention.

Are there any model selection criteria used in the data fitting?

We used the minimum number of phases necessary to fit the data without considering any prior model.

7. Line 204, I think Fig 3d should be cited in here instead of 5d?

Thank you for bringing this to our attention. We have fixed this problem.

Reviewer #3 (Remarks to the Author):

The manuscript "Diversity of bacterial small RNAs drives competitive strategies for a mutual chaperone" by Roca et al. deals with and significantly contributes to three important topics in the field of Hfq-mediated gene expression control in bacteria. First, it proposes a robust single-molecule approach to study the relationship between Hfq and its sRNA clients, which brings in information inaccessible to ensemble methods. As probably the most striking example of such new findings, one can now clearly see that sRNAs are capable of prolonged coexistence on the same Hfq hexamer, sometimes with surprisingly high stoichiometries and variable outcomes, which, put in the context of the timing of sRNA-dependent regulation in bacterial cells, will urge the community to reconsider the established view of such mechanisms. This work also enriches and further minimizes the previously proposed division of Hfq-binding sRNAs into classes by adding a critical biochemical component – the way they behave when running against more or less apt competitors for a shared limiting resource (Hfq molecules). This elevates the definition sRNA classes from the original "one sRNA – Hfq – one target" paradigm to the more physiologically relevant "many sRNAs – Hfq – many

targets” perspective, somewhat where the field is now clearly heading. Finally, this study sheds new light on the key phenomenon of the whole story – sRNA cycling on Hfq. Albeit known since already quite a long time and actively studied by several research groups, it remains elusive from the point of view of its driving forces and the extent to which we can predict the outcome of competition between two given sRNAs.

Similarly to earlier studies, the authors come to the conclusion that bulk thermodynamic or kinetic parameters of individual sRNA-Hfq interactions cannot explain why some sRNAs compete for Hfq more successfully than others. By contrast, the topology of each sRNA-Hfq binding and the extent to which sRNAs are ‘flexible’ with regard to the choice of Hfq faces they engage seem to hold the key to this riddle. Specifically, the authors propose that clashing on the same Hfq surface will typically drive inevitable and fast sRNA exchange, whereas any possibility to occupy separate niches on the same Hfq hexamer will favour prolonged co-residence of several sRNAs (soft competition). At the same time, the first scenario is complicated by the poorer access of a competitor to the already occupied site, whereas the second looks like a sure way to slowly push the resident out by using unoccupied sites as a toehold for invasion. These somewhat opposed tendencies may indeed explain a good deal of what we currently know about the competition properties of class I and class II sRNAs. I also foresee that both the method and the proposed paradigm will stimulate similar research in other molecular systems involving binding of multiple RNAs to a shared protein.

While convinced by the data, I find that the authors may still improve their presentation by better explaining their analysis to non-specialists, providing details about certain controls, and clarifying or further supporting ambiguous wording/interpretations, as outlined in the following (minor) points.

1. It could be useful to better introduce the meaning of the multiple values in Supplementary tables. Why are they multiple in the first place? (One could probably add column subheadings like “ $\tau_{\text{bind_diss1}}$ ”, “ $\tau_{\text{bind_diss2}}$ ”, “ $\tau_{\text{bind_diss3}}$ ” and explain that they come from fitting two or three exponentials.) The term ‘amplitude’ may also be misunderstood by non-specialists (e.g., as ‘amplitude of the fluorescent signal’ instead of ‘proportion of molecules showing the specified behaviour’).

These are very helpful suggestions. We have revised Supplementary Tables 2-4 and added some additional information in the table legend. We also substituted the term “fraction” for “amplitude” in the tables, Fig. 5, and Supplementary Figs. 6, 7 and 8 (new Supplementary Figs. 7, 8 and 9).

2. One technical point is the actual number of C-terminal tails per hexamer which are immobilised on the slide. This may be an important parameter, given that their movement significantly contributes to sRNA cycling on Hfq. Although the authors took multiple precautions to minimize the number of attachment-competent tails, do they have an idea about the actual Hfq/AviHfq ratio in their BioHfq prep? This could be done by SDS-PAGE or by western blotting.

This is an important point and we thank the reviewer for his suggestions. We performed additional controls, which are shown in the new Supplementary Figure 3. Please see the response to Reviewer 2.

3. Another technical issue that deserves a more detailed treatment is how the authors ensured that the spots they observed and quantified always corresponded to single Hfq hexamers and not to two (or more) closely located, microscopically unresolved complexes. This is obviously important in the context of the proposed multiple co-binding of sRNAs. Judging by the density of spots (Supplementary Fig. 3) and the high frequency of observed co-residency events (certainly higher than would be expected from by-chance colocalization; Fig. 4, Supplementary Fig. 8), I do not think that this factor may significantly confound the data and affect their interpretation. But I feel that this issue should be properly discussed in the Methods section to exclude possible doubts.

To deal with this issue, we adjusted the sample concentration to have a low density of molecules (~500 molecules per FOV). Additionally, we manually inspected individual areas of interest (AOIs) to ensure only one molecule was present within this area of the slide. AOIs with two or more neighboring molecules were discarded from the analysis. We have clarified this point in the Methods (lines 543-545 and 548-550).

4. I have a problem with term ‘strength’ as it is employed in the section “Strength of RNA-Hfq interactions predicts active competition” and in Fig. 5. What does it mean? It clearly is not the dissociation constant (the values reported for these sRNAs in literature are actually fairly close to each other and do not follow the order proposed in Fig. 5). To me this notion has more to do with the number of possible high-affinity contacts between each sRNA and different Hfq sites (as illustrated in Supplementary Fig. 4). This is not strength in a common biochemical sense (i.e., bulk affinity measured by an ensemble approach, without distinction of the underlying enthalpic and entropic contributions

from different binding sites). I think it is important to be careful about wording in this section to avoid the already challenged interpretation that individual affinities (Kd) determine the outcome of sRNA competition for Hfq.

Thank you very much for your clarification and suggestion. We agree that the term “strength” doesn’t precisely capture our meaning. As also pointed out by Reviewer 1 (question 1), the number of RNA·Hfq contacts may not predict thermodynamic affinity. We have revised the manuscript to clarify that we are referring to the number of nucleotides in the Hfq binding motifs (line 266 (title of section), lines 272-275, lines 280-282, line 299 (on discussion), Fig. 5, Fig. 6). Please see our response to Reviewer 1.

5. Fig. 3d and the corresponding main text: I was not aware that ChiX interacts with the rim. Could the authors comment on this?

ChiX contains an internal U-rich motif that can interact with the rim (light blue box in Fig. 3d). However, we also think this rim interaction is favored by the adjacent AAN motif that anchors the RNA on the distal face. To avoid the connotation of independent binding to the rim, we revised the text (line 208) to read that the truncated ChiX is unable to reach the proximal face or rim.

6. To the same panel: I would replace “Resident left after competition” with “Resident remaining after competition” to avoid misunderstanding.

Fig 3 has been revised as suggested.

7. Inspired by the same panel and the last section of Results: in the case of ChiX_dist, does it show more soft competition than longer ChiX versions, or it loses the ability to displace DsrA altogether?

Thank you for this astute suggestion. We performed additional analysis on the data for this truncated RNA, and have added a new supplementary figure (Supplementary Fig. 11) showing that the percentage of stable coexistence with DsrA is significantly larger for ChiX_dist than with ChiX and ChiXΔtail. This important result provides additional evidence that stable coexistence increases when the RNAs do not overlap around Hfq. We discuss it in the revised results’ section “Interactions with opposite Hfq faces encourage coexistence” (lines 304-307). We thank the reviewer for leading us to it.

Conversely, would a ChiX variant without the distal site-binding motif still be capable of soft competition?

We did not determine this, but we hypothesize that it will behave similarly to DsrA in complex with any other class I sRNA. That is, the percentage of stable coexistence will not be as significant (Fig. 5, middle).

8. L. 150: should not it be “60% of CX events”, given the 15% initial burst mentioned in Supplementary Table 2?

This is correct. We have revised it in line 149.

9. LI. 171-172: the difference between displacing and all binding events (Supplementary Fig. 6) seems to be small. Is it significant (e.g., by K-S test)?

As suggested, we have performed K-S tests for all the data presented in Supplementary Fig. 6 (new Supplementary Fig. 7). For this, we used Physics: Tools for Science (College of St Benedict, St John’s University) at [KS-test Data Entry \(csbsju.edu\)](https://www.csbsju.edu/ks-test-data-entry). Indeed, the results report no significant difference between all binding events and events leading to displacement. In response, we removed this panel from Fig. 2. We argue that binding leading to displacement, as with all other types of binding, is mainly affected by the presence of a bulky resident on the protein. We have edited our manuscript to incorporate these revised results (Fig. 2 and new Supplementary Fig. 7, lines 166-173, 192-193 and Statistical analysis section in Methods). We thank the reviewer for helping us clarify this issue.

10. L. 204: should be “Figs. 1b, 1d and 3d”.

We have made this correction. Thank you.

11. LI. 241-244: the observation that the homotypic complexes exist longer than the heterotypic ones (70 s vs 40 s) seems to be paradoxical. Do the authors have an idea why it may be so?

We believe that stable coexistence of sRNAs in the same face of Hfq can only occur for very tight and favorable RNA configurations on Hfq, while for sRNAs positioned in opposite faces this condition could be more relaxed, resulting in shorter lifetimes for heterotypic complexes. This is an interesting point so we have added a comment about it in lines 253-256.

12. L. 260-262: unless the authors have actual data or simulation results to support this assertion, I would tone it down to “Among them, RydC can be expected/seems to make the fewest number of interactions...”

This statement was based on a low-resolution crystal structure. However, the revised wording in lines 273-274 is softer and based on a predicted ranking of Hfq-binding motifs.

13. Throughout the text: a better wording would be ‘model/function is fitted into data’ and not the other way around.

We have revised this throughout.

14. L. 611: This reference lacks further authors (E. Massé is not the only one on the paper).

Corrected. Thank you.

15. L. 618: ‘performance’.

We have revised this. Thank you.

16. Fig. 2f seems to be identical to Supplementary Fig. 6d.

Yes, they were the same in the original manuscript. The plot was used in Fig. 2 as a representative case and in Supplementary Fig. 6 (new Supplementary Fig. 7) for completeness. We have omitted panel f from Fig. 2 based on your observation in point 9, and as a result, duplication of panels is no longer a problem.

17. L. 754 (Fig. 4c): the sentence “Additional sRNAs disrupt stable complexes” is not completely accurate since the coexistence of 2 or 3 sRNAs on Hfq seems to be equally likely.

We agree with this observation and have revised this sentence in the legend to Fig 4.

18. L. 762 (Fig. 5): there are no representative trajectories here.

Thank you for noticing this; it referred to an older draft of Fig. 5. We have revised the legend to Fig. 5.

19. Supplementary Fig. 1: A supporting reference is needed for the statement “We chose this sRNA-mRNA regulatory pair...”

We have added a reference for this in Supplementary Fig. 1.

Alexandre Smirnov

REVIEWERS' COMMENTS

Reviewer #1 (Remarks to the Author):

The response letter and the corrections introduced by the Authors into the manuscript fully addressed my concerns. The manuscript has been appropriately improved and is now ready for the publication in Nature Communications.

Reviewer #2 (Remarks to the Author):

The authors have addressed my previous comments. The revised manuscript reads nicely and easier to understand. I recommend the publication of the manuscript. I do have one very minor suggestion: `t_bind_diss` is better labeled directly in Figure 2b (right panel), and Figure S7a can be removed.

Reviewer #3 (Remarks to the Author):

The authors have fully addressed all the points raised by the reviewers. I thank the authors for this excellent revision and, in particular, for the additional data they have now provided to support their message. I also find that the presentation of the rather complex results of this important study has considerably improved: it will certainly speak to broader readership. With this I strongly recommend the publication of this paper.

Alexandre Smirnov

REVIEWER COMMENTS

Reviewer #1 (Remarks to the Author):

The response letter and the corrections introduced by the Authors into the manuscript fully addressed my concerns. The manuscript has been appropriately improved and is now ready for the publication in Nature Communications.

Thank you.

Reviewer #2 (Remarks to the Author):

The authors have addressed my previous comments. The revised manuscript reads nicely and easier to understand. I recommend the publication of the manuscript. I do have one very minor suggestion: `t_bind_diss` is better labeled directly in Figure 2b (right panel), and Figure S7a can be removed.

Thank you for your further suggestions. We have now defined `t_bind_diss` in Fig. 2a, right and removed Supplementary Fig. 7a.

Reviewer #3 (Remarks to the Author):

The authors have fully addressed all the points raised by the reviewers. I thank the authors for this excellent revision and, in particular, for the additional data they have now provided to support their message. I also find that the presentation of the rather complex results of this important study has considerably improved: it will certainly speak to broader readership. With this I strongly recommend the publication of this paper.

Alexandre Smirnov

Thank you for your kind comments and also thank you for such a thorough review.